# WASH interventions and child diarrhea at the interface of climate and socioeconomic position in Bangladesh

Pearl Anne Ante-Testard [1] ✉, Francois Rerolle[1,2], Anna T. Nguyen[3], Sania Ashraf[4], Sarker Masud Parvez[4,5], Abu Mohammed Naser[6], Tarik Benmarhnia[2], Mahbubur Rahman [4], Stephen P. Luby [7], Jade Benjamin-Chung [3,8] & Benjamin F. Arnold [1]

Many diarrhea-causing pathogens are climate-sensitive, and populations with the lowest socioeconomic position (SEP) are often most vulnerable to climate-related transmission. Household Water, Sanitation, and Handwashing (WASH) interventions constitute one potential effective strategy to reduce child diarrhea, especially among low-income households. Capitalizing on a cluster randomized trial population (360 clusters, 4941 children with 8440 measurements) in rural Bangladesh, one of the world's most climate-sensitive regions, we show that improved WASH substantially reduces diarrhea risk with largest benefits among children with lowest SEP and during the monsoon season. We extrapolated trial results to rural Bangladesh regions using high-resolution geospatial layers to identify areas most likely to benefit. Scaling up a similar intervention could prevent an estimated 734 (95% CI 385, 1085) cases per 1000 children per month during the seasonal monsoon, with marked regional heterogeneities. Here, we show how to extend large-scale trials to inform WASH strategies among climate-sensitive and low-income populations.

Diarrhea is a leading cause of mortality in children under 5 years, accounting for 370,000 deaths in children in 2019[1]. Diarrhea-related deaths among children under 5 years are highest in South Asia and sub-Saharan Africa[2], and improved water, sanitation, and handwashing (WASH) interventions have been identified as a key strategy to reduce diarrhea morbidity and mortality among young children[3,4].

The interface between climate-related diarrhea and socioeconomic position may have profound effects on the impact of household WASH interventions in vulnerable populations. Children born in rural Bangladesh experience high rates of diarrhea and extreme weather events such as heavy rainfall during the annual monsoon[5]. Interventions to help reduce inequalities in diarrheal outcomes following extreme weather are crucial and urgent. Previous studies from the WASH Benefits Bangladesh trial demonstrated that improved WASH interventions significantly reduced child diarrhea[6], with largest reductions in high precipitation periods during the seasonal monsoon[7]. In addition, longer term follow-up of the control and sanitation arms further showed sustained reductions in child diarrhea for more than 3 years[8]. Previous studies using data from the WASH Benefits Bangladesh trial assessed the impact of WASH interventions on child diarrhea during the monsoon season[7,8], but they did not examine the effect of WASH by a granular measurement of

[1]Francis I. Proctor Foundation and Department of Ophthalmology, University of California, San Francisco, San Francisco, CA, USA. [2]Scripps Institution of Oceanography, University of California, San Diego, San Diego, CA, USA. [3]Department of Epidemiology and Population Health, Stanford University, Stanford, CA, USA. [4]Environmental Health and WASH, Health System and Population Studies Division, icddr,b, Dhaka 1212, Bangladesh. [5]Child Health Research Centre, The University of Queensland, South Brisbane, QLD, Australia. [6]Division of Epidemiology, Biostatistics, and Environmental Health, School of Public Health, University of Memphis, Memphis, TN, USA. [7]Division of Infectious Diseases and Geographic Medicine, Stanford University, Stanford, CA, USA. [8]Chan Zuckerberg Biohub, San Francisco, CA 94158, USA. ✉e-mail: pearl.ante@ucsf.edu

socioeconomic position, nor did they assess potential joint interactions between socioeconomic position and season. Health benefits from WASH interventions might be difficult to achieve and even sustain among the households with the lowest socioeconomic position. Households with high socioeconomic position may tend to have more time and resources for activities that improve health and sanitation, which can reduce diarrhea prevalence such as cleaning their residential compound, washing up before and after cooking, handwashing with soap and clean water, and many others. Moreover, resilience of communities against extreme weather events may depend on their social position in society—wealthier households could provide a built and enabling environment to "bounce back" rapidly[9,10].

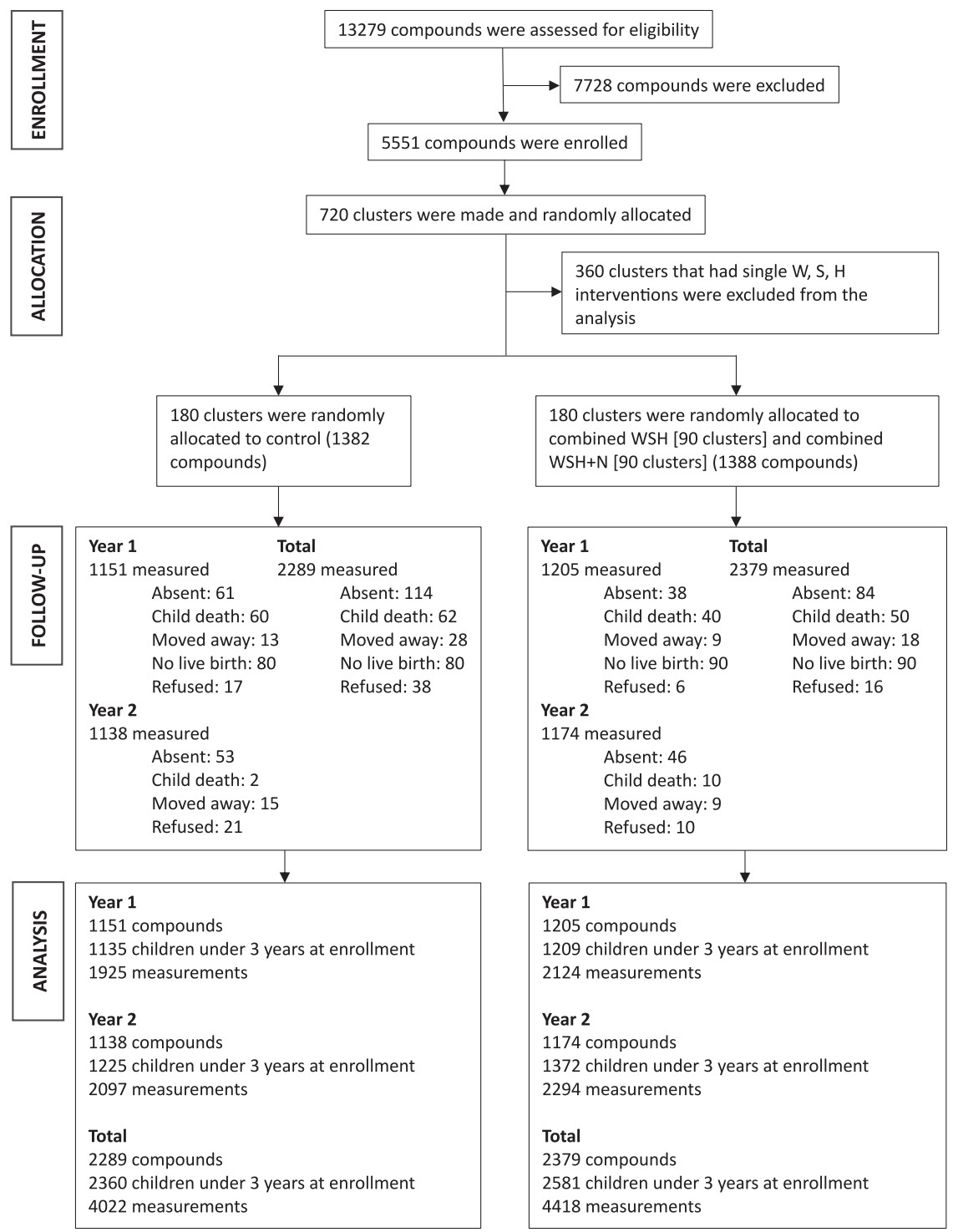

**Fig. 1 | Study participant flow.** Summary of clusters enrolled and randomized to combined Water, Sanitation, Handwashing (WSH) and Nutrition (WSH + N) versus control in the WASH Benefits Bangladesh trial. Intervention groups randomized to single WASH and nutrition interventions were not included in this pre-specified, subgroup analysis. No clusters dropped out of the trial, and follow-up summarizes study compounds over the two-year study period. The total number of children and measurements considered in the analysis was determined following the removal of missing data for diarrhea, exclusion of children aged over 3 years at enrollment, and exclusion of single WASH and nutrition interventions in surveys 1 and 2.

**Table 1 | Baseline characteristics of control and WASH intervention groups included in this analysis of the WASH Benefits Bangladesh trial**

| Baseline characteristics | Control (N = 773) | Intervention (N = 835) |
|---|---|---|
| **Mother's characteristics** | | |
| Mother's age in years (Mean, SD) | 23.5 (4.80) | 24.1 (5.03) |
| **Observations per household and compound** | | |
| Persons per household (Mean, SD) | 4.85 (2.55) | 4.90 (2.37) |
| Persons per compound (Mean, SD) | 14.6 (6.65) | 15.5 (7.42) |
| **Asset-based characteristics included in the construction of the wealth index** | | |
| Land owned in acres (Mean, SD) | 0.125 (0.184) | 0.119 (0.154) |
| Improved wall material (wood, brick, thin) | 557 (72.1 %) | 624 (74.7) |
| Improved floor material (wood concrete) | 74 (9.6%) | 67 (8.0%) |
| Household has electricity | 430 (55.6%) | 465 (55.7%) |
| Household has refrigerator | 53 (6.9%) | 55 (6.6%) |
| Household has bicycle | 203 (26.3%) | 237 (28.4%) |
| Household has motorcycle | 52 (6.7%) | 47 (5.6%) |
| Household has sewing machine | 41 (5.3%) | 45 (5.4%) |
| Has black and white or colored TV | 188 (24.3%) | 193 (23.1) |
| Has one or more wardrobe | 113 (14.6%) | 118 (14.1%) |
| Has one or more table | 536 (69.3%) | 598 (71.6%) |
| Has one or more chair | 560 (72.4%) | 576 (69.0%) |
| Has one or more khat (type of bed) | 455 (58.9%) | 499 (59.8%) |
| Has one or more chouki (type of chair) | 602 (77.9%) | 649 (77.7%) |
| Has one or more mobile | 650 (84.1%) | 688 (82.4%) |
| **Asset-based characteristics not included in the construction of the wealth index** | | |
| Primary water source: shallow tubewell | 550 (71.2%) | 633 (75.8%) |
| Store drinking water | 381 (49.3%) | 381 (45.6%) |
| Reported treating water today/ tomorrow | 4 (0.5%) | 0 (0%) |
| Own their latrine | 330 (42.7%) | 346 (41.4%) |
| Latrine has concrete slab | 692 (89.5%) | 723 (86.6%) |
| Latrine has functional water seal | 168 (21.7%) | 156 (18.7%) |
| No visible feces on floor of latrine | 327 (42.3%) | 315 (37.7%) |
| Has a potty for defecation | 54 (7.0%) | 60 (7.2%) |
| Primary handwashing location has water/soap | 146 (18.9%) | 163 (19.5%) |
| Household has radio | 34 (4.4%) | 29 (3.5%) |
| Has one or more clock | 249 (32.2%) | 275 (32.9%) |

The intervention group includes children born into clusters that received a combined water, sanitation, and handwashing (WSH) intervention either alone or in combination with nutritional supplementation (WSH + N), with details in Methods.
SD standard deviation, N total number.

**Table 2 | Summary of effect modifiers by intervention group**

| Effect modifier (baseline) | Control (N = 773) | WASH (N = 835) |
|---|---|---|
| Wealth scores (Median, Q1 to Q3) | 1.29 (0.76 to 1.88) | 1.34 (0.76 to 1.88) |
| **Effect modifier (surveys 1 and 2)** | **Control (N = 4022)** | **WASH (N = 4418)** |
| Season: Dry | 2120 (52.7%) | 2339 (52.9%) |
| Monsoon | 1902 (47.3%) | 2079 (47.1%) |

Wealth scores were derived using a principal component analysis. Monsoon season was defined as the weeks with elevated precipitation (May 27–September 27, 2014, and April 1–September 26, 2015), which were based on weekly precipitation data matched to the study cohort. Dry season included the other dates.
Q1 quartile 1, Q3 quartile 3, N total number.

position could benefit less due to lower ability to respond to intervention or could benefit more due to higher background risk and thus more potential to gain from intervention. We then sought to extrapolate the trial results to estimate the potential cases averted under an efficacious WASH intervention in rural Bangladesh by combining effect estimates by key effect modifiers (monsoon, wealth) with national-extent spatial layers and geostatistical models. We generate fine-scale estimates of regions throughout rural Bangladesh most likely to benefit from combined WASH interventions to reduce the climate-related diarrhea burden, accounting for spatial variation in socioeconomic position, geographic setting (excluded urban areas) and population density of children under 3 years. In doing so, we demonstrate how to leverage large-scale, randomized controlled trials to identify populations beyond the trial that could benefit most from interventions to reduce climate-sensitive diseases.

## Results
### Study population and characteristics
The trial created and randomly allocated 720 clusters and enrolled 5551 pregnant women in 5551 compounds to an intervention or control group. Previous studies within the WASH Benefits Bangladesh trial used data from the 720 clusters from seven arms including the double-sized control (Fig. 1) to answer questions related to the early impacts of WASH on diarrhea[6,8,12]. In this pre-specified analysis, we focused on the combined WSH, combined WSH + Nutrition (WSH + N) and double-sized control arms to ensure statistical power for effect modification analyses while maintaining a consistent WASH package in the intervention group. There was a total of 360 clusters that were randomly allocated to combined WASH interventions (i.e., WSH and WSH + N; hereafter, WASH intervention) and control groups, with 8440 diarrhea measurements from 4941 children in surveys 1 and 2 (Fig. 1). Intervention and control groups were balanced across socio-demographic and household asset-based characteristics (Table 1).

In this study, we used a principal component analysis-derived wealth index[13,14] and season as possible effect modifiers of the WASH intervention on child diarrhea. A participant's wealth was mostly defined by asset-based ownership such as land ownership and improved wall material, and access to goods such as electricity (Table 1, Supplementary Table 1). Monsoon season was defined by the dates with elevated precipitation (May 27–September 27 in 2014 and April 1–September 26 in 2015)[7]. To gain a better understanding of the climate characteristics during these defined monsoon dates, we used monthly precipitation, maximum temperature and soil moisture data from TerraClimate[15] to characterize climate conditions during the trial. Monthly means of key climate variables were relatively higher during the monsoon season (Supplementary Fig. 1). The distribution of wealth index at baseline as well as the distribution of children with diarrhea

Here, we conducted a pre-specified secondary analysis[11] of the WASH Benefits Bangladesh trial with a particular focus on the combined WASH intervention's ability to reduce climate-related diarrhea risk among children along a gradient of socioeconomic position. Based on prior results, we hypothesized that reductions in diarrhea during the monsoon might differ along a socioeconomic gradient–households with lower socioeconomic

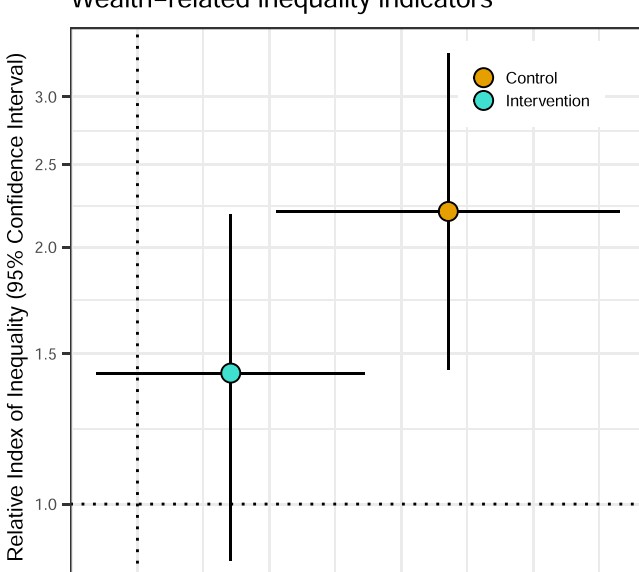

## Wealth–related inequality indicators

**Fig. 2 | Relative Index of Inequality and Slope Index of Inequality between the control and intervention groups.** RII (*y*-axis) and SII (*x*-axis) are the two major measures in epidemiological studies for quantification of inequalities in relative and absolute scales, respectively. These are regression-based indicators which use all subgroups compared to a pairwise comparison that ignores other groups. To estimate these indices, we first ranked the individuals from those with the lowest socioeconomic position (rank = 0) to those with the highest socioeconomic position (rank = 1) in the cumulative distribution of the wealth index. We used a generalized linear model using a binomial family with a log link. We calculated the RII as the ratio of the value at the bottom of the socioeconomic position (intercept) to the value at the top (intercept + slope). Meanwhile, the SII is the difference between the value at the bottom of the socioeconomic position (intercept) and the value at the top (intercept + slope). RII = 1 and SII = 0 indicate no inequality. RII > 1 and SII > 0 indicate inequality disfavoring the participants with the lowest socioeconomic position (i.e., child diarrhea is more concentrated among the participants with the lowest socioeconomic position). The central estimates as depicted by circles represent the RII (*y*-axis) and SII (*x*-axis). Error bars represent 95% confidence intervals. Please refer to Table 2 for the sample size details (*n* = 8440 observations).

measurements obtained during the dry and monsoon seasons in surveys 1 and 2 were balanced between control and intervention groups (Table 2).

## Wealth-related inequalities in child diarrhea between control and intervention arms

We measured wealth-related inequalities in child diarrhea between the two arms at the relative and absolute scales by calculating the Relative Index of Inequality (RII) and Slope Index of Inequality (SII), respectively. The RII is the ratio of the predicted outcomes between the populations with the lowest and highest socioeconomic position in the wealth distribution, while the SII is the difference[16]. There were strong relative and absolute inequalities in child diarrhea favoring wealthier households in the control group (Fig. 2). On the relative scale, the participants with the lowest socioeconomic position were 2.2 times (95% Confidence Interval 1.4, 3.4) more likely to report child diarrhea than the wealthiest participants. On the absolute scale, the households with the lowest socioeconomic position reported around 5 percentage points (95% CI 2%, 7%) higher diarrhea prevalence than the wealthiest. However, we found that the RII and SII estimates among the group that received WASH intervention were lower [RII: 1.4, 95% CI (0.9, 2.2); SII: around 1 percentage point (95% CI −0.6%, 3%)] than the control.

## Effects of WASH interventions by socioeconomic position

We assessed the effect of the WASH intervention along a continuous wealth index score and tertiles of the wealth index in pre-specified analyses (Fig. 3). There was heterogeneity across asset-based characteristics that make up the wealth index with increasing mean values from the households with lowest socioeconomic position to the wealthiest households (Supplementary Table 1). We observed a wealth gradient of increasing diarrhea prevalence from the participants with the highest socioeconomic position to those with the lowest socioeconomic position in the control group (Fig. 3). Reductions in diarrhea due to WASH were largest in the bottom third of the wealth distribution or the wealth tertile with the lowest socioeconomic position (additive interaction *p*-value = 0.07, multiplicative interaction *p*-value = 0.22)− with diarrhea prevalence of 8.1% (95% CI 6.4%, 9.8%) in control versus 4.5% (95% CI 3.3%, 5.8%) in WASH [prevalence difference = 3.6% (95% CI 1.4%, 5.7%), prevalence ratio = 1.8 (95% CI 1.3, 2.4)].

## Effect modification by monsoon season

Bangladesh's monsoon integrates several correlated shifts in temperature, precipitation, and soil moisture that co-occur from approximately May through September (Supplementary Fig. 1). We thus used monsoon season as a climate-related effect modifier for the trial as a pragmatic measure that combines several correlated variables that could influence diarrhea-pathogen transmission and has a clear, actionable interpretation. The seasonal monsoon was a strong modifier of the efficacy of the WASH intervention on diarrhea, with nearly all diarrhea reduction during the monsoon season (additive interaction *p*-value = 0.0003, multiplicative interaction *p*-value = 0.001). During the monsoon season, diarrhea prevalence among children in the control group was 8.3% (95% CI 6.6%, 10.0%) and among children with improved WASH prevalence was 4.3% (95% CI 3.4%, 5.2%) [prevalence difference = 4.0% (95% CI 2.2%, 5.9%), prevalence ratio = 1.9 (95% CI 1.5, 2.5)] (Fig. 4) which is consistent with previous studies that conducted an in-depth analysis of climate-related effect modifiers[7] and monsoon season as effect modifier[7,8].

## Joint effect modification by socioeconomic position and monsoon season

Reductions in diarrhea due to WASH intervention were largest among the tertile with the lowest socioeconomic position during the monsoon season (diarrhea prevalence of 10.3% (95% CI 7.6%, 13.0%) in control versus 4.6% (95% CI 3.1%, 6.0%) in WASH [prevalence difference = 5.7% (95% CI 2.7%, 8.6%), prevalence ratio = 2.2 (95% CI 1.5, 3.3)]), although the joint interaction was not statistically significant at conventional levels due to limited sample sizes within the large number of strata (additive interaction *p*-value = 0.95, multiplicative interaction *p*-value = 0.82) (Fig. 5). This pattern was consistent when we estimated diarrhea prevalence and effect estimates along the continuous wealth index using splines (Supplementary Fig. 2).

## Identifying populations throughout rural Bangladesh most likely to benefit from the WASH intervention

We combined intervention trial effects estimated by wealth index during the monsoon season with national surfaces of wealth[17] and population[18] to extrapolate effects from the trial to a broader population conditional on variables[19] including socioeconomic position, monsoon, age (<3 years), geographic setting (excluded urban areas) and population density (excluded grid-cells with less than or equal 2 children under 3 years) with a goal of identifying regions of rural Bangladesh that could benefit most from an efficacious WASH to reduce climate-related diarrhea among young children (Fig. 6). The national wealth surface generated from WorldPop was based on the 2011 Demographic and Health Survey (DHS) wealth index in Bangladesh and was constructed by taking the principal component of a group of household living standards and characteristics, which is

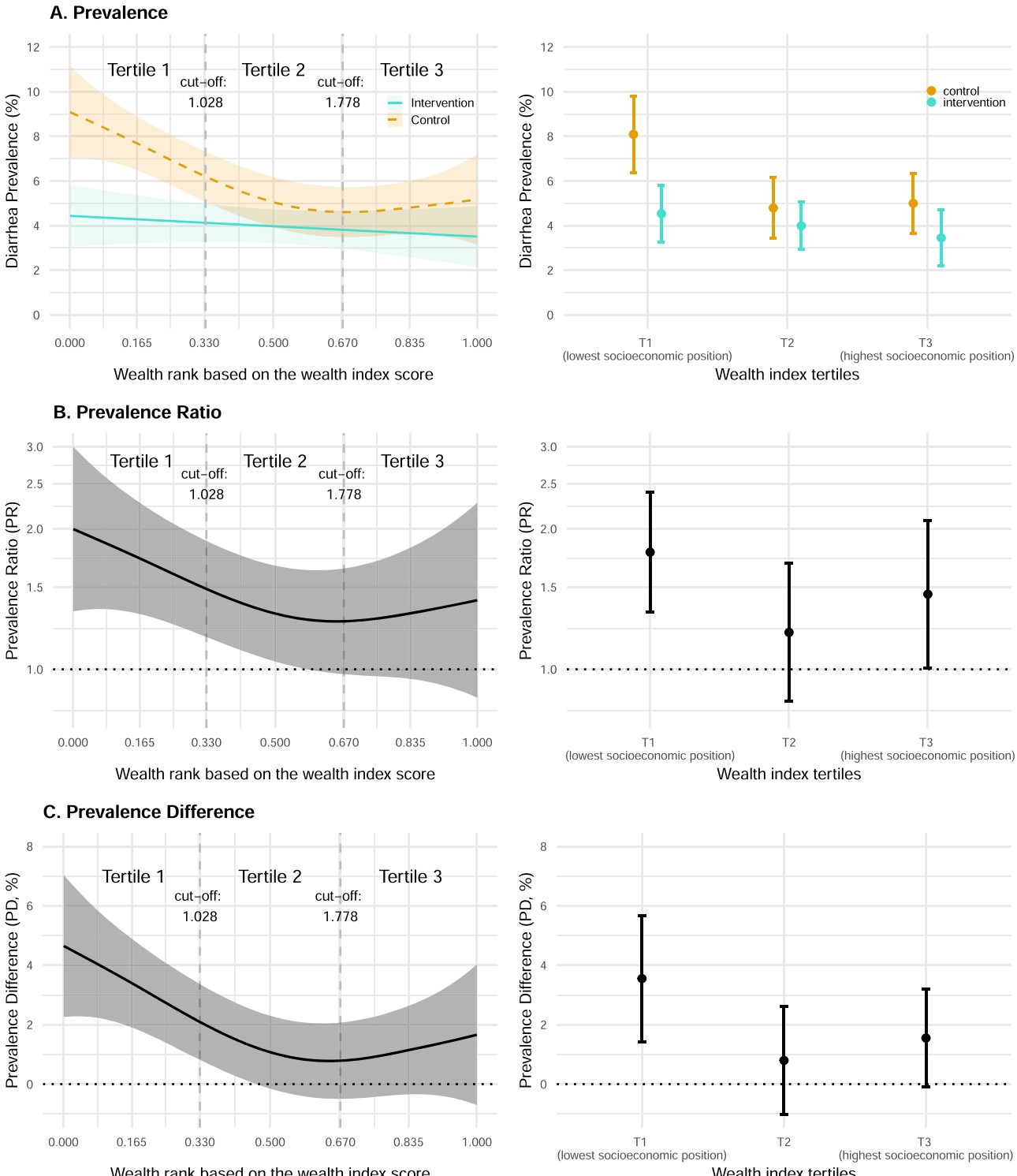

**Fig. 3 | Effect of WASH interventions by socioeconomic position.** Left panels show estimates using a continuous wealth score. Right panels show estimates using tertiles of wealth index, a pre-specified grouping of the continuous score. Sample sizes by group and wealth category are reported in Table 2. **A** Diarrhea prevalence along the wealth distribution and wealth tertiles for children <3 years at enrollment in the control and intervention groups. The central estimates as depicted by splines and circles represent the diarrhea prevalence. Shaded areas and error bars represent 95% confidence intervals. **B** Prevalence ratio of child diarrhea along the wealth distribution and wealth tertiles between the control and intervention groups. The Y-axis is on a log scale. The central estimates as depicted by a spline and circles represent the prevalence ratios. Shaded area and error bars represent 95% confidence intervals. **C** Prevalence difference of child diarrhea along the wealth distribution and wealth tertiles between the control and intervention groups. The central estimates as depicted by a spline and circles represent the prevalence differences. Shaded area and error bars represent 95% confidence intervals. **A**–**C** Please refer to Table 2 for the sample size details (*n* = 8440 observations).

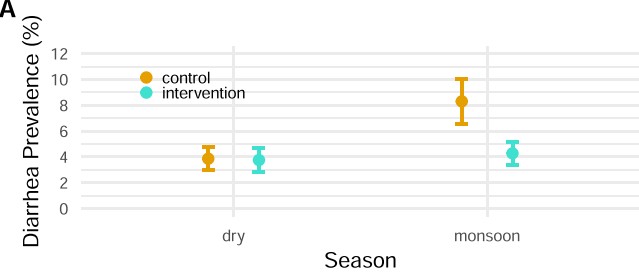

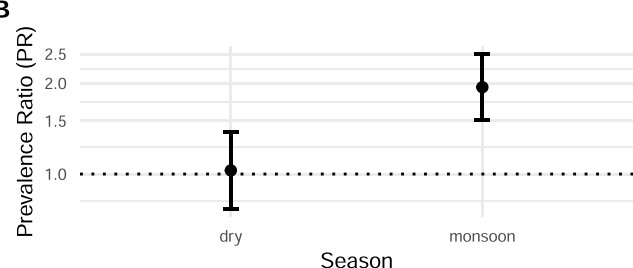

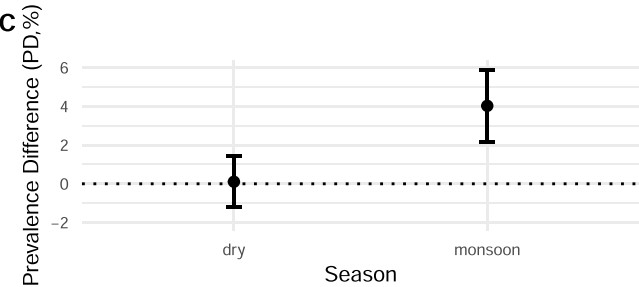

**Fig. 4 | Effect of WASH interventions by season. A** Diarrhea prevalence during the dry and monsoon seasons in the control and intervention groups. Monsoon season was defined as weeks with elevated precipitation (May 27–September 27, 2014, and April 1–September 26, 2015). These dates were based on weekly precipitation data matched to the study cohort. Dry season were the other dates. The central estimates as depicted by circles represent the diarrhea prevalence. Error bars represent 95% confidence intervals. **B** Prevalence ratio of child diarrhea during the dry and monsoon seasons between the control and intervention groups. The Y-axis is on a log scale. The central estimates as depicted by circles represent the prevalence ratios. Error bars represent 95% confidence intervals. **C** Prevalence difference of child diarrhea during the dry and monsoon seasons between the control and intervention groups. The central estimates as depicted by circles represent the prevalence differences. Error bars represent 95% confidence intervals. **A–C** Please refer to Table 2 for the sample size (*n* = 8440 observations).

similar to the wealth index we used in the main analysis. To evaluate the similarity between the WorldPop-projected wealth index layer and the observed wealth index from the WASH Benefits Bangladesh trial, we calculated the Spearman correlation and produced a scatter plot (Supplementary Fig. 3). The correlation between wealth measures was 0.39 (using wealth rank) and 0.43 (using wealth scores), suggesting a moderate degree of correlation, and of similar magnitude to other comparisons of national-level, projected wealth surfaces with directly measured wealth indices[20]. In our analysis, we adopted a more conservative approach by utilizing the wealth rank. However, it is important to highlight that variations exist in the construction of the wealth indices. The WorldPop wealth index, derived from the DHS, integrated WASH-related asset-based variables in its construction, whereas we opted to exclude them when constructing the wealth index for the WASH Benefits Bangladesh trial. Hence, it is essential to exercise caution when interpreting the projections in this context.

The WorldPop wealth surface was based on a Bayesian model-based geostatistics (see details in the Methods section) in combination with high resolution gridded spatial covariates and aggregated mobile phone data applied to GPS-located household survey data on poverty[17]. Overall, the projected diarrhea cases prevented during monsoon season by an efficacious WASH intervention was 734 (95% CI 385, 1085) cases per 1000 children <3 years per month across rural Bangladesh, with marked heterogeneity by district. The populations predicted to benefit most from improved household WASH based on diarrhea cases prevented were in north-central and north-west Bangladesh (Fig. 6c, d), reflecting larger diarrhea reductions among the households with lowest socioeconomic position during the monsoon (Fig. 5c) and the high density of children living in low socioeconomic position, rural households in that region.

## Sensitivity analyses

To assess the robustness of the findings when using the wealth index, we conducted a sensitivity analysis using maternal education defined by the years of education. Our results were similar when using maternal education as a measure of socioeconomic position instead of the wealth index, illustrating the close relationship between household wealth and maternal education (Supplementary Table 2, Supplementary Figs. 4 and 5).

A previous analysis of this trial found that WASH reduced diarrhea most during periods with at least one day of heavy rain in the previous week (>80th percentile rainfall) under a 1-week lag[7]. We conducted a sensitivity analysis to assess joint effect modification by socioeconomic position and heavy rainfall, which showed overall consistent findings with the main analysis that stratified by wealth and season (Supplementary Fig. 6). Although the joint interaction was also not statistically significant (additive interaction *p*-value = 0.15, multiplicative interaction *p*-value = 0.27), we observed largest reductions in child diarrhea among the tertile with lowest socioeconomic position during heavy rain (diarrhea prevalence of 10.5% (95% CI 7.5%, 13.4%) in control versus 4.8% (95% CI 3.3%, 6.4%) in WASH [prevalence difference = 5.7% (95% CI 2.1%, 9.2%), prevalence ratio = 2.2 (95% CI 1.5, 3.2)]) (Supplementary Fig. 6).

Furthermore, we projected the diarrhea cases prevented by wealth across the entire year due to an efficacious WASH was 339 (95% CI 105, 573) cases per 1000 children <3 years per month. This analysis yielded results consistent with the primary findings, revealing similar geographical patterns to those observed in the main analysis concentrated on the monsoon period. Notably, the regions in north-central and north-west Bangladesh emerged as the regions that would benefit the most, although the overall effect size was slightly reduced when averaged over the whole year (Supplementary Fig. 7).

## Discussion

This pre-specified secondary analysis of a cluster-randomized trial demonstrated socioeconomic inequalities in child diarrhea that were reduced by a household WASH intervention. Improved household WASH reduced child diarrhea most among young children with the lowest socioeconomic position and nearly all reductions in diarrhea risk were during monsoon season, when climatically driven pathogen transmission lead to diarrhea prevalence doubling in the control group but no commensurate increase among children in the intervention group across all socioeconomic levels. When extrapolating the intervention trial estimates conditional on key effect modifiers, we found that the populations predicted to benefit most from an efficacious household WASH intervention in terms of diarrhea cases prevented during the monsoon were in north-central and north-west Bangladesh —regions with areas of lower socioeconomic position, reflecting larger diarrhea reductions among the households with the lowest socioeconomic position.

This study demonstrates how effect estimates from existing randomized trials can be extrapolated beyond the original study population to help inform climate mitigation strategies in

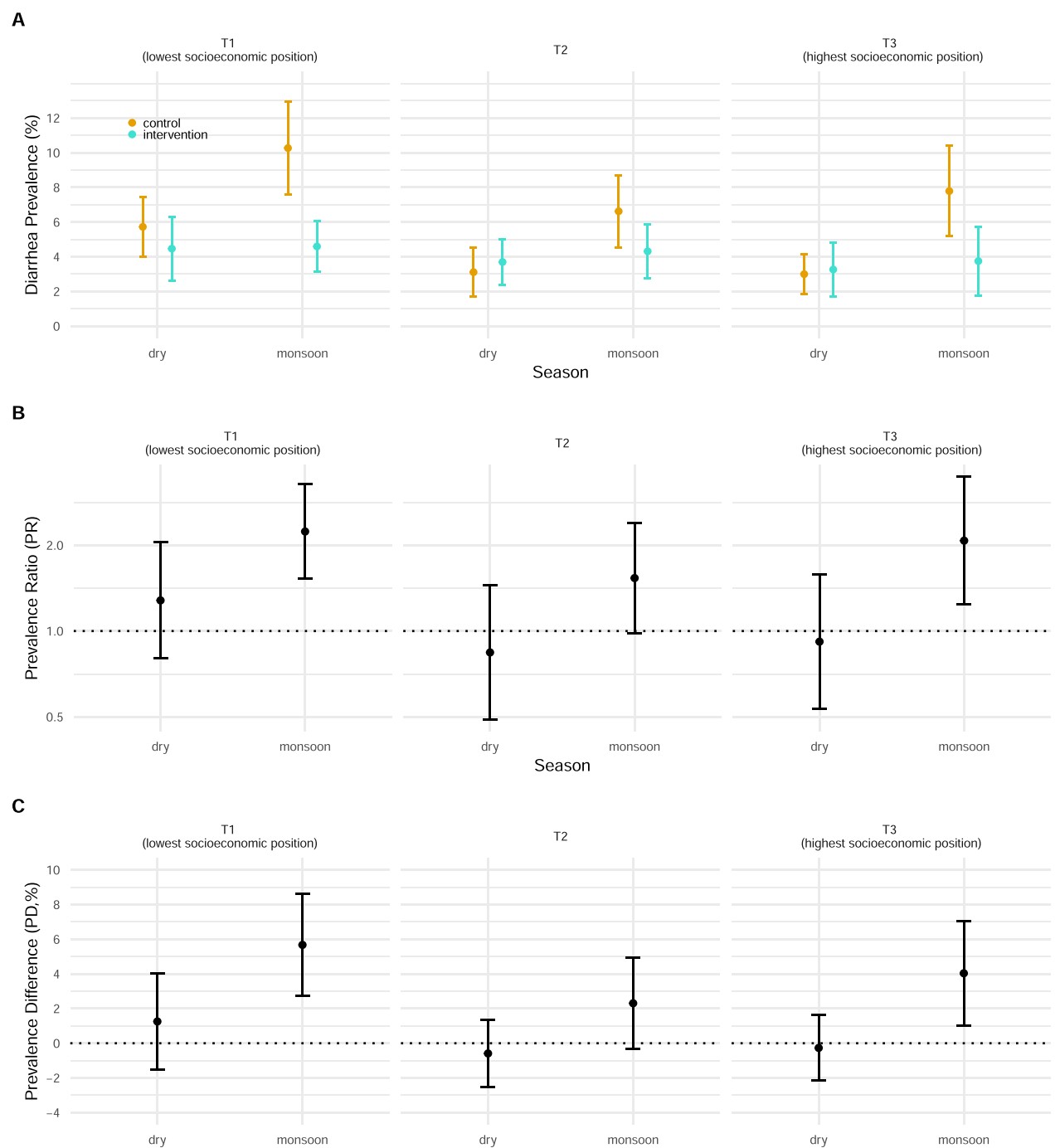

**Fig. 5 | Effect of WASH intervention jointly by socioeconomic position and season. A** Diarrhea prevalence along the tertiles of wealth index during the dry and monsoon seasons in the control and intervention groups. The central estimates as depicted by circles represent the diarrhea prevalence. Error bars represent 95% confidence intervals. **B** Prevalence ratio of child diarrhea along the tertiles of wealth index during the dry and monsoon seasons between the control and intervention groups. The *Y*-axis is on a log scale. The central estimates as depicted by circles represent the prevalence ratios. Error bars represent 95% confidence intervals. **C** Prevalence difference of child diarrhea along the tertiles of wealth index during the dry and monsoon seasons between the control and intervention groups. The central estimates as depicted by circles represent the prevalence differences. Error bars represent 95% confidence intervals. **A**–**C** Please refer to Table 2 for the sample size details (*n* = 8440 observations).

vulnerable populations[21]. Randomized controlled trials generate effect estimates with high internal validity but those effects are rarely generalized to a larger target population[22]. The results herein demonstrate how large trials can be reexamined to identify key effect modifiers related to climate-equity and then combined with novel, high-resolution data layers to identify populations that would benefit most from climate mitigation strategies. Trials as large and expensive as WASH Benefits Bangladesh may not be repeated for decades, if ever. Innovative extensions of existing cluster randomized trials to inform climate mitigation strategies provide timely evidence in the face of rapid acceleration in climate change, without waiting years or decades

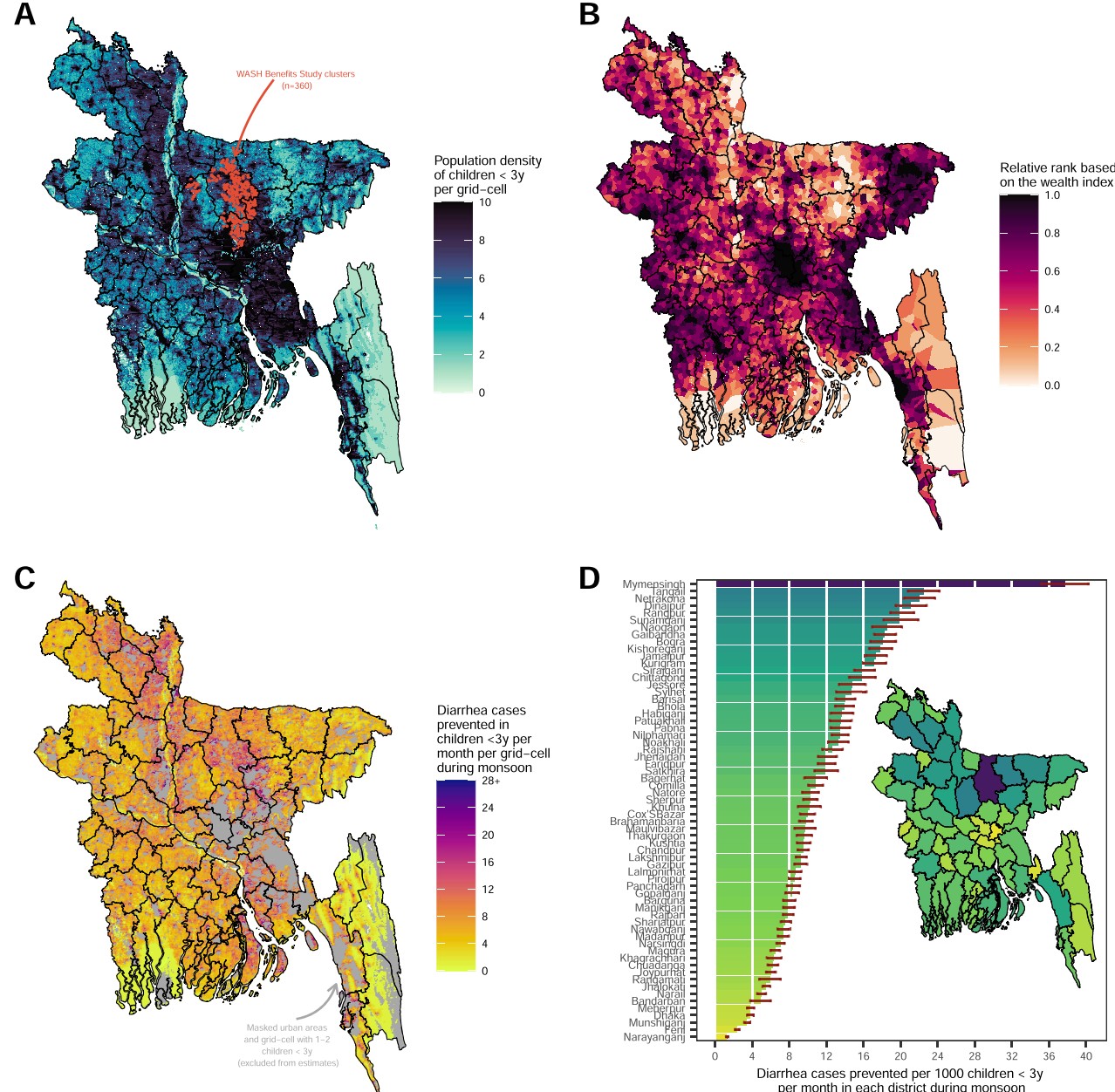

**Fig. 6 | Diarrhea cases prevented by a combined WASH intervention along a continuous wealth gradient during the seasonal monsoon per grid-cell at a 1-kilometer resolution and per district in rural Bangladesh. A** WASH Benefits Study clusters that were included in the analysis ($n = 360$, 3981 child measurements) and gridded population density of children <3 years estimated from WorldPop. **B** Distribution of relative wealth rank based on the Demographic and Health Surveys wealth index with Bayesian model-based geostatistics and high resolution gridded spatial covariates and aggregated phone data per grid-cell at 1-km resolution estimated from WorldPop. **C** The projected diarrhea cases prevented by combined WASH by wealth during monsoon season per month in children <3 years per grid-cell at 1-km resolution. Masked urban areas defined by the Global Human Settlement and grid-cells with less than or equal to 2 children <3 years. **D** The projected diarrhea cases prevented by combined WASH per 1000 children <3 years per month in panel (**C**) aggregated for each administrative district ($n = 64$ districts). The bars represent the sum of the grid-cell projected diarrhea cases prevented by combined WASH during monsoon season in children <3 years per month at the district level. Error bars represent 95% confidence intervals.

to field new trials. A caveat of the approach is that it requires sufficiently large trials with enough relevant variation in key effect modifiers to estimate reliable effects among crucial subgroups (here: wealth and season). Another caveat is that effects projected from efficacy trials assume the intervention could be implemented at a very high level with high uptake at even larger scale, a strong assumption given the challenge of translating trial interventions to real-world programs[23]. Despite these caveats, identifying populations most likely to benefit from improved

WASH using a climate-sensitive, equity lens (Fig. 6) should still be informative for public health programs.

This study provides evidence of the importance of assessing the interface between socioeconomic position and climate with inequalities and climate change being two of the main barriers in WASH[24]. Our results show that among children in the control group, disparities existed by socioeconomic position and season and that the WASH intervention reduced or even nullified existing disparities. Although the joint interaction between wealth and

season was not statistically significant at conventional levels, we observed that the tertile with the lowest socioeconomic position still had higher diarrhea prevalence than the wealthiest tertile in the control group during the monsoon season. The resilience, adaptation mechanism, and access to healthcare may be different for people with the lowest and highest socioeconomic position when it comes to mitigating higher diarrhea risk during monsoon season. Households with the lowest socioeconomic position were found to be less resilient when faced with unexpected events such as natural calamities and pandemics[9,10]. It is possible that without an intervention such as WASH, households with the lowest socioeconomic position may be unable to adapt to and show resiliency from the impacts of diarrhea during monsoon season compared to wealthier households, especially as such inequalities may be exacerbated in the context of climate change.

Identifying and targeting the most vulnerable populations is a crucial step in leveraging the benefits of WASH and in reaching the populations that need these interventions the most. A systematic mapping study found that most WASH interventions lacked social inclusion or intersectional considerations in their intervention designs[25]. This could be a current gap for many WASH trials and this study is one of the efforts to addressing it. Although our findings show that the WASH intervention reduced disparities by socioeconomic position and season, consciously incorporating a social equity lens in the design and planning of future trials and programs is paramount. Without careful attention to equity from planning to implementation, it is possible that these interventions may unintentionally generate or widen inequalities between different groups (i.e., so-called intervention-generated inequalities)[26–28]. This may happen when careful attention is not given to individuals with lower socioeconomic position and people with higher socioeconomic position benefit at an accelerated rate from an intervention. This could also be a potential explanation why the middle wealth tertile is lagging behind the tertiles with the highest and lowest socioeconomic position (Fig. 3)–not enough attention is given to this group in comparison to the other two groups at the extreme ends of the wealth gradient. Interventions that are delivered similarly to all recipients may result to differential outcomes because the individuals with the lowest socioeconomic position or less educated individuals are less able to access, understand and engage with the intervention[28]. This shows that implementing WASH that does not generate inequalities would require a tailored approach[29]. Members of our team have previously assessed equity in the WASH Benefits Bangladesh interventions with respect to measures of adherence[30] and others have assessed barriers and inequalities in effective WASH practices in the African setting[24,31]. These previous studies have demonstrated improved equity in WASH practices and this present analysis demonstrates that those improvements in equity carry forward to improvements in equity in child diarrhea.

Previous analyses of the WASH Benefits Bangladesh trial explored effect modification by season, but this analysis builds from and extends the previous work in several ways. Nguyen et al. centered their analysis on precipitation and temperature as effect modifiers, characterizing the monsoon as a composite measure of specific climate variables[7]–in this analysis we thus used the monsoon season as a measure of seasonal changes in climate indicators since it integrates correlated changes in temperature and precipitation and simplifies the interpretation. Another recent summary from the same trial population focused on the longer-term effect of only the sanitation intervention on child diarrhea by monsoon season between one and 3.5 years after the intervention[8]. Here, we focused on the combined WSH and WSH + N interventions and measured its effect on child diarrhea one- and two-years post-implementation. Lastly, we have presented

the joint effect modification by socioeconomic position and season when assessing the effect of WASH on child diarrhea, an extension beyond considering season alone as an effect modifier.

Our analysis had some limitations. We relied on a caregiver-reported outcome which might have been subject to social desirability bias and courtesy bias that may lead to underreporting of the outcome and misclassification. However, findings from Luby et al. showed no evidence of misclassification after a negative control analysis[6]. Additional evidence that these findings were not primarily influenced by courtesy bias is the lower prevalence of identified enteropathogens in the intervention households based on a previous analysis[32]. We also used the wealth index as our measure of socioeconomic position. While it is a reliable measure of wealth in the Global South, it can only represent relative wealth within a population[33]–in this case rural Bangladesh, which may limit comparisons across populations. For this reason, we conducted a sensitivity analysis using a continuous measure of maternal education that showed consistent findings with the wealth index. Another limitation was that the trial was conducted only in areas that were not prone to being heavily flooded during seasonal monsoon which might have underestimated or overestimated the effects of WASH on diarrhea during the monsoon season in more flood prone areas. Moreover, although we have conditioned on some variables and key effect modifiers when extrapolating the trial estimates, it is possible that we have unmeasured confounding in the target population where we extrapolated the study trial effects to and may have missed other potential effect modifiers that are important. Another limitation is that the diarrhea prevalence was overall quite low in the study, as low as 4% in the intervention group, which could have limited the study's inference of effect modification if there was not enough diarrhea to have an effect on in some data stratum (Fig. 5). As a corollary, effect size could also vary less in areas with higher diarrhea prevalence. Furthermore, in this paper, we did not consider the various enteric pathogens and illnesses responsible for diarrhea, such as cholera, shigellosis, and rotavirus. These pathogens might exhibit distinct mechanisms and varying sensitivities to climate[34–38]. The higher impact of WASH during the monsoon season on the risk of diarrhea might be attributed to the distinct impact of WASH on seasonally varying pathogens[32,39], whose transmission may not be solely driven by climatic conditions. Nevertheless, a study revealed that cholera and shigellosis, both identified as climate-sensitive, exhibited comparable relationships with climate variables despite differences in disease epidemiology[37]. A final caveat is the extrapolation of cases prevented by an efficacious WASH, which may not have fully taken into consideration the complexity of the interaction between diarrheal diseases, the monsoon season, and their inter-annual variations since it relied on estimates from only two monsoon seasons. The use of generalized additive models (GAMs) captured non-linear relationships between continuous wealth, improved WASH, and child diarrhea during the monsoon season, but data from two years may not have adequately encompassed the full scope of climate variability and inter-annual fluctuations, as well as the complex, non-linear dynamics of pathogen dynamics that cause diarrheal disease. Future studies over longer time periods could provide additional evidence for varying efficacy of the WASH interventions, by wealth and season, though randomized controlled trials of WASH interventions rarely extend for more than two years. Future meta-analyses that pool results over many different trials could help capture effect heterogeneity over a wider range of seasonally varying climate conditions.

Despite the limitations, this study had several strengths. This study provides additional knowledge on the effects of WASH on child diarrhea and provides an example on how to extrapolate trial estimates to a broader population at the interface of climate and socioeconomic position–making it possible to identify vulnerable populations that would benefit the most from these interventions. This study is also a randomized controlled trial, which offers high internal

validity. We also made sure that our intervention group satisfy the consistency assumption in causal inference[40], in addition to the assumptions of exchangeability and positivity that randomized controlled trials provide.

In conclusion, we highlight the capacity of WASH Benefits intervention to improve population resilience to climate-related diarrhea—by reducing the wealth disparity in diarrhea, with largest reductions in diarrhea amongst the children with the lowest socioeconomic position during the monsoon season. The study demonstrates how to assess equity of intervention effects and extrapolate effects from trials to help target programs to populations who would benefit most.

## Methods
### Study data
We conducted a secondary analysis of the WASH Benefits Bangladesh cluster randomized controlled trial. The study design and rationale[41] and the primary outcomes[6] of this trial have been previously published. There was no data collection software used in this pre-specified secondary analysis.

Children aged less than 3 years at enrollment living in the compound were eligible for the caregiver-reported diarrhea. The analysis focused on index children in the birth cohort, and other children living within the same compound that were younger than 3 years at the time of study enrollment. Children with missing outcome data were excluded. Our analysis only focused on survey rounds 1 and 2 (2014 and 2015, respectively).

The trial was conducted in rural communities in Gazipur, Kishoreganj, Mymesingh and Tangail districts. These districts are in central rural Bangladesh where the main source of livelihood of the population is agriculture. Between May 31, 2012, and July 7, 2013, the trial enrolled pregnant women who were identified during the community-based surveys who were expected to deliver in the 6 months following enrollment. Written informed consent was obtained from participants prior to enrollment.

Compounds were enrolled within 720 geographically matched clusters with eight clusters per matched block. Within a matched block, the trial randomly allocated eight clusters to receive: improved water (W), improved sanitation (S), improved handwashing (H), improved nutrition (N), combined WSH, combined WSH + N, and a double-sized control arm. This study focused on children enrolled in control clusters and clusters that received the combined WSH intervention. Within each geographically matched block, the present analysis included four clusters: 2 controls, 1 WSH, and 1 WSH + N. The combined WSH clusters were compared with the control clusters.

Participants and the data collectors were not masked to intervention assignment due to the nature of the interventions. However, the data collection and intervention teams were different individuals. The results were unmasked after the primary outcome analyses were completed.

### Intervention group definition
The WASH Benefits Bangladesh trial comprised seven arms: water (W), sanitation (S), handwashing (H), nutrition (N), combined WSH, combined WSH + N, and double-sized control. The water intervention encompassed the use of an insulated storage container for drinking water, along with the utilization of Aquatabs. Sanitation efforts involved the implementation of a sani-scoop, potty, and an improved double-pit pour flush latrine. Handwashing interventions included the provision of a handwashing station, a storage bottle for soapy water, and laundry detergent sachets for the preparation of soapy water. In addition, the nutrition arm entailed the use of lipid nutrient supplements (LNS) between 6 and 24 months, along with a storage container for LNS, exclusive breastfeeding until 180 days, and the introduction of complementary food at 6 months.

Here, we focused on the combined WSH, combined WSH + N and double-sized control arms to ensure statistical power. The intervention arm included the combined WSH and combined WSH + N arms. We excluded single arms to ensure a consistent WASH package in the intervention group[40]. We found no evidence for any added benefit of combined WSH + N with respect to combined WSH in diarrhea[6].

### Outcome
The outcome of interest was the caregiver-reported diarrhea in the past 7 days among children enrolled in the trial. We defined diarrhea as having at least three or more episodes of loose or watery stools in 24 h or the latest one stool with blood based on caregiver-reported symptoms in the past 7 days. This variable was binary—0 as non-event and 1 as having the event.

### Defining the effect modifiers
**Socioeconomic position.** The wealth index was the main socioeconomic position indicator to measure wealth. It is an asset-based composite measure of wealth based on a set of household assets and characteristics. In constructing the wealth index, we used a principal component analysis of asset-based variables measured for all participants at enrollment (Supplementary Table 1)[13]. We excluded all WASH-related variables as recommended by UNICEF[14]. We excluded asset-based variables with near-to-zero variation (i.e., owning a radio and improved roof) and high levels of missingness (at least 10%, i.e., owning a clock)[42,43]. Missing values for continuous variables were replaced by the mean[13], and for missing factor levels, another level was created. We used continuous wealth index scores in the GAMs and three quantiles (i.e., tertiles) to average over more observations to improve statistical power in the effect modification analyses. We have pre-specified tertiles compared to quintiles or quartiles after a review of the number of children that would be analyzed in each subgroup without consideration of the outcome or effect. Tertiles ensured that there was adequate statistical power when conducting the subgroup analyses, while still allowing for us to assess a pattern from lower to higher socioeconomic position. For the continuous wealth index scores, we specifically used the relative wealth rank of the participants in the cumulative distribution of the wealth index score which is a continuous variable ranging from 0 to 1. Based on the factor loadings, some of the asset-based variables that contribute the most to the wealth variation are having one or more 'khat' or bed, electricity in the household, owning a TV and having one or more chair. Meanwhile, the presence of a 'chouki,' a specific type of stool, demonstrates a negative factor loading, suggesting that wealthier households tend to have Western-style chairs, whereas the utilization of 'choukis' is linked to manual labor involving squatting. Another possible explanation could be the likelihood that wealthier households simply prefer chairs over 'choukis.'

**Monsoon season.** We defined monsoon season dates using weekly precipitation data from the Multi-Source Weighted-Ensemble Precipitation from the GloH20[44] matched to the study cohort[7,38]. Monsoon season was marked by the weeks with elevated precipitation and persistent rainfall, where the rolling 5-day average was above 10 millimeters (May 27–September 27 in 2014 and April 1–September 26 in 2015) based on previous analyses of the trial[7,38]. Meanwhile, dry season included other weeks. This was a binary variable—0 was coded for the dry season and 1 for the monsoon season. We calculated the monthly mean of key climate variables during the trial to characterize the monsoon versus dry seasons in the study.

### Geospatial layers
WorldPop provides a high-resolution map of the total number of people per grid-cell at a 1-km resolution in Bangladesh in 2014[18]. We estimated and mapped the total number of children under 3 years by

first obtaining the proportion of those between 0 and 14 years (0.31 based on the World Bank)[45] and then multiplying it by 0.21 (or $\frac{3}{14}$). We assumed a uniform age distribution within the 0-14 range, thus employing the proportion of 0.21.

The wealth index layer was also obtained from WorldPop. Specifically, we obtained the 2011 estimates of the mean DHS wealth index score per grid square. It is based on a Bayesian model-based geostatistics in combination with high-resolution gridded spatial covariates and aggregated mobile phone data applied to GPS-located household survey data on poverty from the DHS Program[17]. We then estimated and mapped the relative wealth rank per grid square based on the WorldPop wealth layer.

The urban and rural layer was obtained from the Global Human Settlement[46]. We used this layer to mask the non-urban areas from the analysis. The shapefiles were obtained from GADM[47].

## Statistical analyses

The analysis was by intention-to-treat. First, we conducted descriptive statistics of the baseline characteristics of the children's mothers and asset-based household assets. We then conducted descriptive statistics of the effect modifiers used in the analyses—wealth index scores (at baseline) and season (surveys 1 and 2).

Second, we measured socioeconomic inequalities in child diarrhea by calculating the RII and SII to measure inequalities at the relative and absolute scales, respectively[16]. These are regression-based indicators commonly used to measure inequalities.

Third, we estimated the effects of WASH interventions by socioeconomic position, season and jointly by socioeconomic position and season. The general approach that we used to estimate the effects of combined WASH on diarrhea, with wealth tertiles and season as effect modifiers, is using a generalized linear model (GLM) using a binomial family with an identity link for the absolute effects and log link for the relative effects. We used robust standard errors to account for clustering at the block level. We estimated the 95% confidence intervals of the effect estimates using a linear combination of regression coefficients.

In assessing the effect of WASH on diarrhea by the continuous wealth score, we fitted a generalized additive model (GAM) to capture any non-linear relationships[48] between continuous wealth index and child diarrhea. We used a non-parametric smoother, specifically the penalized cubic regression spline to avoid overfitting[49] and fit using the restricted maximum likelihood[50]. Block-level random effects were included in the model to account for clustering of observations at the block level (which was the level of matched randomization). This model allowed for the relationship between socioeconomic position and diarrhea to vary by control and combined WASH group (reference group). We fitted GAM[51] using a Gaussian family with identity link to estimate the prevalence difference, and using a binomial family with log link to estimate the prevalence ratio together with their 95% confidence intervals using the tidymv package[52].

Fourth, we assessed the effect modification of WASH by socioeconomic position, season and jointly by socioeconomic position and season. We primarily assessed effect modification on the additive scale, which is a measure that is more relevant in public health[53]. We additionally assessed effect modification on the multiplicative scale. We assessed effect modification by comparing the models with and without the interaction term through a Wald-type F test to test for statistical significance (Supplementary Text 1).

## Projecting the impact of an efficacious WASH intervention across Bangladesh to estimate preventable burden

Lastly, we projected diarrhea cases for children under 3 years prevented by the WASH intervention conditional on socioeconomic position, monsoon, age (<3 years), geographic setting (excluded urban areas) and population density (grid-cells with less than or equal to 2 children below 3 years were excluded) throughout rural Bangladesh by combining

intervention trial effects from splines estimated across a wealth gradient during the monsoon season with national surfaces of wealth and population. First, we used the WorldPop-projected wealth index layer based on the 2011 DHS wealth index. Second, we used WorldPop population layer to estimate the number of children under 3 years in each grid-cell at a 1-kilometer resolution. Third, we restricted the projections to rural areas of Bangladesh by masking urban areas defined using the Global Human Settlement. Fourth, we used GAM model prevalence difference estimates between the control and WASH based on wealth index (specifically wealth rank) to project diarrhea cases by integrating steps 1–4. We assumed that the control group reflects the baseline WASH conditions for the projection. We applied the GAM model for diarrhea prevalence difference between control and WASH arms by wealth rank to estimate the number of prevented cases by multiplying it by the proportion of children under 3 years indexed by wealth rank during the monsoon season. We projected effect estimates by continuous wealth rank, age and limited our inference to only rural areas and populations that were similar as the WASH Benefits study population. We projected diarrhea cases prevented per month per grid-cell at a 1-kilometer resolution in rural Bangladesh using this formula:

$$diarrhea\,prevented\,by\,WASH = children\,under\,3y \times PD(control - WASH)_{i.monsoon.rural} \times 4\,weeks,$$

where PD is the diarrhea prevalence difference between control and WASH and $i$ is the wealth rank based on the continuous wealth index scores. We multiplied the estimate by 4 weeks to calculate the diarrhea prevalence prevented per month since diarrhea was measured in the past 7-day period. The estimates assume that 7-day prevalence reflect incident episodes of diarrhea, which accords with a recent, high-resolution longitudinal cohort in Bangladesh that found 1350 out of 1526 (88.5%) of diarrhea episodes from birth to 24 months that lasted less than seven days[54].

We then aggregated the grid-cell estimates at the district level to estimate the diarrhea cases prevented per 1000 children under 3 years per month in each district (i.e., zilas) whilst excluding estimates from urban areas. We defined urban areas based on the Global Human Settlement[46]. We aggregated grid-cell level standard errors estimated from the GAM within each administrative district to calculate the district-level 95% CI.

## Sensitivity analysis

We conducted a sensitivity analysis using maternal education instead of wealth index as a measure of socioeconomic position to estimate diarrhea prevalence, prevalence ratio, and prevalence difference. We also used precipitation data from the Multi-Source Weighted-Ensemble Precipitation from the GloH2O[44]. We created binary variables of rainfall (no heavy rain versus heavy rain) under 1-week lag, which indicates whether there was at least one day in the prior week where total precipitation was above the 80th percentile of all daily totals of precipitation. We considered this indicator based on a previous analysis of the trial that demonstrated strong associations between this indicator and the effect of the WASH intervention on child diarrhea[7]. We assessed the joint effect modification between socioeconomic position and rainfall by comparing the models with and without the interaction term through a Wald-type F test. We also projected the diarrhea cases prevented for children under 3 years throughout rural Bangladesh by wealth averaged over the entire year by combining trial estimates from the GAM that captured non-linear patterns.

## Inclusion and ethics

The protocol of the original study (Clinical Trial Registration NCT01590095) was approved by the Ethical Review Committee at the International Centre for Diarrhoeal Disease Research, Bangladesh (PR-11063), the Committee for the Protection of Human Subjects at the University of California, Berkeley (2011-09-3652), the Institutional

Review Board at Stanford University (25863) and at the University of California, San Francisco (22-36722).

**Reporting summary**

Further information on research design is available in the Nature Portfolio Reporting Summary linked to this article.

## Data availability

The pre-analysis plan, de-identified data and accessible links to the publicly-available spatial and climate data are available through the Open Science Framework (OSF, https://osf.io/xwndg/)[11]. The data do not include the geographic locations of the study clusters to protect the participants' privacy.

## Code availability

All analyses were conducted in R (version 4.2.1 "Funny Looking Kid"). Analysis scripts and instructions to reproduce all analyses are available in the Open Science Framework repository (https://osf.io/xwndg/)[11].

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

## Acknowledgements

The WASH Benefits Bangladesh trial was funded by the Bill & Melinda Gates Foundation (OPPGD759) with additional support for this analysis from the National Institute of Allergy and Infectious Diseases (R01-AI166671 to B.F.A.).

## Author contributions

P.A.A.-T. and B.F.A. developed and drafted the analysis plan with input from A.T.N., S.A., S.M.P., A.M.N., T.B., M.R., S.P.L. and J.B.-C. P.A.A.-T. conducted the analysis with input and guidance from B.F.A. P.A.A.-T. constructed the tables and figures with input from F.R. and B.F.A. F.R. and A.T.N. provided support in the collection of the map layer and climatic variables, respectively. All authors contributed to the interpretation of the results. P.A.A.-T. wrote the initial draft of the manuscript with input and conceptual guidance from B.F.A. All authors contributed to the subsequent revisions. All authors read and approved the manuscript.

## Competing interests

The authors declare no competing interests.
