## [Peer Review File · Nature Communications]

WASH interventions and child diarrhea at the interface of climate and socioeconomic position in BangladeshREVIEWER COMMENTS

Reviewer #1 (Remarks to the Author):

L 138: should let the reader know what are the definitions of wealthiest and poorest—e.g. top and bottom tertiles of... Otherwise, the statements are meaningless.

L 142-145: Best not to say: "the WASH intervention reduced these observed inequalities" because, according to my calculations with just the 2 significant digits, those two point estimates are not statistically very distinguishable.

A major thing missing in this manuscript are confidence intervals around the estimated "cases" prevented. Just giving a point estimate is not very useful. There are CIs at the district level (Fig 6D) but not on the summary numbers.

L 349: They weren't "pair" matched—there were 8 clusters in a block.

L 429-433: For GLM, first it is said that Zou's approach was used to analyze "diarrhea"—I think they mean diarrhea prevalence—which is a binary outcome, so that using the robust estimator at the individual level accounts for the mismatch in distributions (Bernoulli vs. Poisson). At least, that's what is done in Zou's paper. But then, it says a robust estimator was used to account for the cluster level correlation. Which was done? I don't see how both could be done simultaneously. I am worried they may have mistakenly just used the robust estimator at the individual level. If that was not done, and the sandwich variance estimator was calculated at the cluster level, that is OK, it will also account for the Bernoulli-Poisson misspecification. Note that the within-block correlation is ignored in this approach, but that just means that the analysis would be conservative, as the block-level correlation would not be accounted for (just as an unpaired comparison of pair-matched data will be less powerful).

The main design paper for this trial (Arnold 2013) does not seem to have the secondary analyses of wealth and monsoon interaction effects—where were these prespecified? Need a reference.

L 466-468: "We multiplied the estimate by 4 weeks to calculate the diarrhea prevalence prevented per month since diarrhea was measured in the past 7-day period."

This is stated without any rationale at all—and then "diarrhea prevalence prevented" is termed "cases prevented" which also is not justified in the text. To take an extreme example—what if all the diarrhea episodes are persistent diarrhea, i.e. lasting >2wks? $P=I*D$, after all. Can they reference any methodological article that translates diarrhea prevalence into incidence in this manner, where duration is ignored and only the reference period is used? This is the basis for a key part of this manuscript, and needs to be well-justified. I don't recall having seen anything like this.

I would not be surprised if incidence were reduced proportional to the reduction in prevalence, but even this is not a sure bet—many interventions, e.g. rotavirus vaccine, will shift the whole distribution, which means the tails reduce more rapidly, with higher effectiveness for definitions of more severe diarrhea.

Reviewer #2 (Remarks to the Author):

Ante-Testard et al. studied the impact of Household Water, Sanitation, and Handwashing (WASH) interventions, showing that improved WASH is associated with reduced diarrhea disease risk and that the benefit is the largest in those with low socioeconomic status. I have some concerns about their claim on the association with climate. My main comments are below.

- WASH interventions have already been shown to reduce diarrheal incidence in Bangladesh by some of the same authors (e.g., <https://doi.org/10.1371/journal.pmed.1004041>). Addressing how

this work might differ from previous publications needs to be better stated. For instance, it is unclear if the population described under "Study population and characteristics" has been previously used to address some of these questions in already published papers. If that is the case, I think the document needs to acknowledge that this is the same population that has been studied in the past.

- The effect of climate on the transmission of enteric diseases has long been studied in this region. This literature has largely been ignored in the current form of this manuscript. In addition, sensitivity to climate factors tends to be disease-specific, and by looking at diarrhea in general, the authors are oversimplifying the complex dynamics that each disease experiences with climate. Examples of these papers are:

- o Cholera: <https://doi.org/10.1073/pnas.1108438109>
- o Rotavirus: <https://doi.org/10.1073/pnas.1518977113>,
<https://doi.org/10.1073/pnas.1719579115>
- o Shigellosis: <https://doi.org/10.1371/journal.pone.0107223>

- On a similar note, different enteric diseases have different seasonal trends, and their association with temperature or flooding is not always monotonic. The result that the authors present here about a higher impact of WASH during the monsoon season on diarrheal risk might be due to a differential effect of WASH on different pathogens and not necessarily due to climatic conditions per se. Additionally, a previous work has already looked at the role of sanitation and socioeconomic variables on the spatial variation of rotavirus seasonality in Dhaka, finding that drinking water from tube wells, and not socioeconomic factors, has a protective effect against cases during the monsoon (<https://doi.org/10.1093/infdis/jiz436>).

- When the authors write, "which is consistent with a previous in-depth analysis of climate-related effect modifiers." The reference is <https://doi.org/10.1101/2022.09.25.22280229>. "Influence of temperature and precipitation on the effectiveness of water, sanitation, and handwashing interventions against childhood diarrheal disease in rural Bangladesh: a re-analysis of a randomized control trial." How is that previous preprint different from what is being presented here?

- "We estimated that a similar intervention at scale could prevent 734 cases per 1,000 children per month during the seasonal monsoon, with marked heterogeneity by region. The analysis demonstrates how to extend large-scale trials to inform WASH strategies among climate-sensitive and low-income populations." I don't think two years of data is sufficient to make this conclusion. It oversimplifies the effect of climate covariates, their inter-annual variations, and the nonlinear nature of disease transmission.

Reviewer #3 (Remarks to the Author):

- What are the noteworthy results?

This interesting study uses data from a large-scale RCT in Bangladesh to do several things: 1) using pre-specified analysis to analyse the effect of combined WASH and combined WASH+nutrition interventions on reported diarrhoea in the diarrhoea-peak season (the monsoon); 2) using pre-specified analysis to estimate the effects of the intervention by wealth tertile; and 3) model the likely effects of the interventions across the country (in post-hoc analysis?). I would like to see this innovative paper published, provided the comments can be addressed, and I believe Nature Communications to be a relevant and appropriate outlet for it.

- Will the work be of significance to the field and related fields? How does it compare to the established literature? If the work is not original, please provide relevant references.

The work is potentially highly policy relevant, since it presents analysis of the time of year when WASH interventions can be most effective and also suggests that WASH is a highly pro-poor

intervention, benefiting the poorest groups the most. Both are empirical questions that help resolve different competing theories about disease transmission, and the findings presented by the authors are also supported by a recent systematic review and meta-analysis of WASH interventions and child mortality cited by the authors. The paper also presents results of a method to estimate the likely effects of the intervention if it were possible to scale it up to the country level, by applying the estimated relationship between predictive factors and diarrhoea morbidity to geospatial data. The paper presents the implications of the findings in relevant formats, using estimated coefficients and figures to show the relationship between exposures and health inequality. I wondered if it could be clearer what the WASH starting points were for the diarrhoea mapping exercise, and whether there was some way of making this clearer in the figure or associated text. Another implication of the analysis which I thought might be presented more clearly was the sensitivity analysis for the non-linear correlates of diarrhoea disease with respect to by mother's education; specifically, I wondered what the number of years of education was at each inflection point (the maximum and the minimum) and whether either of these were associated with completion of primary education, which itself is potentially policy-relevant since in Bangladesh great efforts have been made to get more girls into secondary education in recent decades which are thought closely related to child health (e.g. https://ieg.worldbankgroup.org/sites/default/files/Data/reports/impact_evaluation_bangladesh_child_health.pdf).

- Does the work support the conclusions and claims, or is additional evidence needed?

I believe the work supports the findings and conclusions drawn, although I have a few queries about the methods and results, as indicated below, which in my view would need to be resolved or addressed before publication.

- Are there any flaws in the data analysis, interpretation and conclusions? - Do these prohibit publication or require revision?

Can the authors respond to the following comments, which I believe should be addressed prior to publication:

- Throughout the paper, the authors refer to 'improved WASH' or the 'WASH intervention'. I think it should be clearer in the paper that the original trial on which this secondary analysis is based provided some aspects of 'improved WASH' but not all - notably I understand there was no intervention to provide improved water supply in quantity, which might itself also have an impact on diarrhoea disease (see Sharma-Waddington et al., 2023, PLOS Med). It is also not clear what the nutrition (N) component of the intervention was, which may be important since the authors argue later in the paper that they ensured consistency of the interventions analysed (combined WASH and WASH+N together), but it is not clear whether the nutrition intervention may have had theoretical or empirical effects on diarrhoea incidence (e.g. directly by providing clean food or indirectly by promoting greater resilience to infection through better nutritional status) over and above that from "improved WASH" alone.

- On the asset index, which is interpreted explicitly by the authors as representing wealth or socioeconomic position, the authors state that they excluded the components of the asset index often included in such indices relating to water and sanitation access, as argued by a UNICEF technical report. I do not disagree, but can it be clearer what the theoretical reason is for excluding these factors from the index. For example, if the reason is that, in a study examining correlates of diarrhoea outcomes, the wealth index should not include any factors which might have a direct effect on the outcome (since, in this analysis, they would not primarily capture variation in the wealth construct, but rather likely disease exposure), then it is not clear why the index also includes quality of flooring, or perhaps electricity or refrigerator, which also might be thought to affect infectious disease exposure in childhood directly. [There is also a policy angle to this, since analysis of the specific amenities available to households that affect their children's disease incidence directly may be policy actionable (e.g. flooring quality, rural electrification), whereas wealth usually isn't seen to be manipulable by policymakers, at least in the short term.] The authors also note that "we excluded asset based variables with near-to-zero variation and high levels of missingness" - can they clarify if this was why the radio and clock variables were excluded from the index? On the interpretation of the loading factors, only one of which is negative for the 'chouki'/stool, is this because households owning choukis were less likely to own chairs?

- When projecting the likely intervention effects country-wide, the authors state that they used WorldPop data based on the 2011 DHS survey wealth index "which is similar to the wealth index we used in the main analysis". If I understand what the authors have done correctly - they are using a method similar to small area estimation used in the poverty mapping literature - one of the assumptions of such analyses is that the variables used in both data sets should be the same or as similar as possible. With regards the wealth measure, it should be clearer how the two measures of wealth differed (presumably the WorldPop includes water and sanitation access, but anything else?), what the implications for (and limitations of) those differences might be. Depending on the answer to this question, the authors may also want to either present results of sensitivity analysis where they re-estimate their results using exactly the same wealth measure as the DHS 2011 index, so presumably including the WASH exposure measures, or re-estimate the WorldPop data using the wealth index used in the author's primary analysis, or at the very least present the loading factors for each index and the correlations across households of the two indices.
- Relatedly, also for the diarrhoea mapping exercise, can it be clearer what specific variables were used to predict diarrhoea by geographical area, and whether these included measures of existing water supplies, sanitation and hygiene exposures? Where the authors state that they only included geographies with "similar geographic patterns observed as the main analysis that focused on the monsoon period", what were these patterns?
- A minor point, where the authors calculated the total number of children under 3 years by multiplying the total number of under-14s by 3/14, is this an accurate calculation based on the expected age distributions of children in Bangladesh?

- Is the methodology sound? Does the work meet the expected standards in your field?

I believe the study uses high quality methods to generate innovative policy-relevant findings, although what was done could be reported more clearly, and more clarity on the differences between the two measures of wealth used in the paper should be given, as indicated above.

- Is there enough detail provided in the methods for the work to be reproduced?

It is generally clear, although what was done for the diarrhoea mapping exercise could be reported more clearly, as indicated above.

Dear Editor and Reviewers,

We would like to express our sincere appreciation for reviewing our manuscript and providing us with the opportunity to improve our work. The constructive and insightful remarks you have offered have been helpful to improve our manuscript, and we hope that our responses adequately addressed your comments and recommendations.

Presented below are our point-by-point responses to the comments made by the Reviewers.

REVIEWER COMMENTS

Reviewer #1 (Remarks to the Author):

L138: should let the reader know what are the definitions of wealthiest and poorest—e.g. top and bottom tertiles of... Otherwise, the statements are meaningless.

Responses: We thank Reviewer #1 for their comment. In calculating the RII and SII, we used the continuous wealth index, specifically the participants' relative rank in the wealth distribution in the regression.

We made the changes to the sentence:

Lines 140-141: *"The RII is the ratio of the predicted outcomes between the wealthiest and poorest populations in the wealth distribution, while the SII is the difference."¹⁵*

L142-145: Best not to say: "the WASH intervention reduced these observed inequalities" because, according to my calculations with just the 2 significant digits, those two point estimates are not statistically very distinguishable.

Responses: Although the 95% CIs for the RIIs and SIIs of the control are overlapping with the 95% CIs of the WASH group, there is still a reduction in the magnitude of the central estimates. However, we also agree that given the overlapping 95% CIs, reduction may not be that evident. In this regard, we rephrased the wordings to "marginally decreased..."

Lines 147-149: *"However, we found that the WASH intervention marginally decreased these observed relative inequalities [RII: 1.4, 95% CI (0.9, 2.2)]; and absolute inequalities [SII: 1 percentage point (95% CI -1% to 4%).]"*

A major thing missing in this manuscript are confidence intervals around the estimated "cases" prevented. Just giving a point estimate is not very useful. There are CIs at the district level (Fig 6D) but not on the summary numbers.

Responses: We agree and in response we calculated the 95% CI for the estimated cases prevented at country level – 734 (95% CI 385, 1085) cases per 1000 children per month during the seasonal monsoon. We added the 95% CI across the manuscript. In addition, we also added the 95% CI for the diarrhea cases prevented by wealth over the entire year in the sensitivity analysis in Lines 237-239 – 339 (95% CI 105, 573) cases prevented.

L 349: They weren't "pair" matched—there were 8 clusters in a block.

Responses: We agree with this comment, and we removed the word 'pair' throughout the manuscript.

L 429-433: For GLM, first it is said that Zou’s approach was used to analyze “diarrhea”—I think they mean diarrhea prevalence—which is a binary outcome, so that using the robust estimator at the individual level accounts for the mismatch in distributions (Bernoulli vs. Poisson). At least, that’s what is done in Zou’s paper. But then, it says a robust estimator was used to account for the cluster level correlation. Which was done? I don’t see how both could be done simultaneously. I am worried they may have mistakenly just used the robust estimator at the individual level. If that was not done, and the sandwich variance estimator was calculated at the cluster level, that is OK, it will also account for the Bernoulli-Poisson misspecification. Note that the within-block correlation is ignored in this approach, but that just means that the analysis would be conservative, as the block-level correlation would not be accounted for (just as an unpaired comparison of pair-matched data will be less powerful).

Responses: We thank Reviewer #1 for this comment and observation. To clarify, we used GLM using a binomial family with identity link for the absolute effects and used robust standard errors to account for clustering at the block level. Meanwhile, we used modified Poisson regression with a robust sandwich estimator for the relative effects and used a generalized estimating equation to account for clustering at the block level. This was motivated since modified Poisson with a robust sandwich estimator was found to be more stable – lesser convergence issues – compared to a log binomial.¹ Nevertheless, in the interest of maintaining consistency and avoiding potential confusion, as suggested by Reviewer #1, we opted for a unified approach, employing GLM for both scenarios—with an identity link for absolute effects and a log link for relative effects for the effect modification by wealth, monsoon season and jointly by wealth and monsoon, and the sensitivity analysis jointly by wealth and heavy rainfall. Despite this modification, the effect estimates and p-values from the Wald test for interaction remained comparable, with negligible differences, and the inference remained unchanged. No issues of convergence were encountered.

We modified this across the manuscript and updated the figures (Fig. 3, Fig. 4, Fig. 5 and Supplementary Fig. 6).

Lines 478-482: *“The general approach that we used to estimate the effects of combined WASH on diarrhea, with wealth tertiles and monsoon season as effect modifiers, is using a generalized linear model (GLM) using a binomial family with an identity link for the absolute effects and log link for the relative effects. We used robust standard errors to account for clustering at the block level.”*

We also modified the R codes for the relative effects:

From using modified Poisson regression with robust sandwich estimator and GEE:

1. Fit a modified Poisson regression with robust sandwich estimator.

Sample codes:

```
geemod_int <- geeglm(diar7d ~ wealth_tertile + Arms +  
  wealth_tertile*Arms,  
  family = poisson(link="log"), id=block, corstr='independence', std.err = "san.se",  
  data = data)
```

We changed to log binomial with robust standard errors:

1. Fit a log binomial.

Sample codes:

¹ Yelland, L. N., Salter, A. B. & Ryan, P. Performance of the Modified Poisson Regression Approach for Estimating Relative Risks From Clustered Prospective Data. *Am. J. Epidemiol.* **174**, 984–992 (2011).

```
glm_int_log <- glm(diar7df ~ wealth_tertile + Arms + wealth_tertile*Arms,  
  family = binomial(link="log"),  
  data = data)
```

2. Get the variance-covariance matrix:

Sample codes:

```
glmfitlog_robust_vcov <- vcov(glm_int_log, cluster=data$block)
```

3. Update the regression fit with the robust standard errors.

Sample codes:

```
glmfitlog_lin_robustse <- coeftest(glm_int_log, vcov. = glmfitlog_robust_vcov)
```

We also changed our approach when estimating the RIIs and SIIs from modified Poisson regression to GLM using a binomial family with log link (while accounting for clustering at the block level using robust standard errors). We revised Fig. 2 and edited the caption.

Caption lines 796-800: *“We used a generalized linear model using a binomial family with a log link. We calculated the RII as the ratio of the value at the bottom of the socioeconomic position (intercept) to the value at the top (intercept + slope). Meanwhile, the SII is the difference between the value at the bottom of the socioeconomic position (intercept) and the value at the top (intercept + slope).”*

For more details, our codes are available on GitHub and mirrored on OSF: <https://osf.io/xwndg/>. We added these details under the section Code Availability.

The main design paper for this trial (Arnold 2013) does not seem to have the secondary analyses of wealth and monsoon interaction effects—where were these prespecified? Need a reference.

Responses: The original trial did not specify this analysis in 2013 but all of the analyses presented in this paper were pre-specified in a pre-registered Statistical Analysis Plan that we completed before starting with this analysis.

We added a citation to line 93: *“Here, we conducted a pre-specified secondary analysis¹¹ of the WASH Benefits Bangladesh trial with a particular focus on the combined WASH intervention’s ability to reduce climate-related diarrhea risk among children along a gradient of socioeconomic position.”*

Reference:

11. Ante-Testard, P. A. & Arnold, B. WASH interventions and child diarrhea at the interface of climate and socioeconomic position in Bangladesh: analysis plan, source data and code. (2022) doi:10.17605/OSF.IO/XWNDG.

L 466-468: *“We multiplied the estimate by 4 weeks to calculate the diarrhea prevalence prevented per month since diarrhea was measured in the past 7-day period.”*

This is stated without any rationale at all—and then “diarrhea prevalence prevented” is termed “cases prevented” which also is not justified in the text. To take an extreme example—what if all the diarrhea episodes are persistent diarrhea, i.e. lasting >2wks? $P=I*D$, after all. Can they reference any methodological article that translates diarrhea prevalence into incidence in this manner, where duration is ignored and only the reference period is used? This is the basis for a key part of this manuscript, and needs to be well-justified. I don’t recall having seen anything like this.

I would not be surprised if incidence were reduced proportional to the reduction in prevalence, but even this is not a sure bet—many interventions, e.g. rotavirus vaccine, will shift the whole

distribution, which means the tails reduce more rapidly, with higher effectiveness for definitions of more severe diarrhea.

Responses: Thank you for this comment. We assumed that the cases of diarrhea in children predominantly lasted less than one week, so that a prevalence measure with a 7-day recall would represent incident episodes. We felt this was a reasonable assumption based on previous, detailed estimates from MAL-ED Bangladesh that tracked episode duration for 1566 diarrhea episodes and found that 88.5% of the episodes were short-duration and acute, lasting for less than seven days.

We added these sentences in the Methods section:

Lines 527-530: *“The estimates assume that 7-day prevalence reflect incident episodes of diarrhea, which accords with a recent, high-resolution longitudinal cohort in Bangladesh that found 1350 out of 1566 (88.5%) of diarrhea episodes from birth to 24 months that lasted less than seven days.”⁵⁵*

Added this reference:

55. Platts-Mills, J. A. et al. Pathogen-specific burdens of community diarrhoea in developing countries: a multisite birth cohort study (MAL-ED). *Lancet Glob. Health* 3, e564–e575 (2015).

Reviewer #2 (Remarks to the Author):

Ante-Testard et al. studied the impact of Household Water, Sanitation, and Handwashing (WASH) interventions, showing that improved WASH is associated with reduced diarrhea disease risk and that the benefit is the largest in those with low socioeconomic status. I have some concerns about their claim on the association with climate. My main comments are below.

- WASH interventions have already been shown to reduce diarrheal incidence in Bangladesh by some of the same authors (e.g., <https://doi.org/10.1371/journal.pmed.1004041>). Addressing how this work might differ from previous publications needs to be better stated. For instance, it is unclear if the population described under “Study population and characteristics” has been previously used to address some of these questions in already published papers. If that is the case, I think the document needs to acknowledge that this is the same population that has been studied in the past.

Responses: We thank Reviewer #2 for their comment. We agree and made the connection to a previous work in this trial more explicit:

Lines 76-80: *“Previous studies from the WASH Benefits Bangladesh trial demonstrated that improved WASH interventions significantly reduced child diarrhea⁶, with largest reductions in high precipitation periods during the seasonal monsoon.⁷ Additionally, longer term follow-up of the control and sanitation arms further showed sustained reductions in child diarrhea for more than 3 years.⁸”*

- The effect of climate on the transmission of enteric diseases has long been studied in this region. This literature has largely been ignored in the current form of this manuscript. In addition, sensitivity to climate factors tends to be disease-specific, and by looking at diarrhea in general, the authors are oversimplifying the complex dynamics that each disease experiences with climate. Examples of these papers are:

o Cholera: <https://doi.org/10.1073/pnas.1108438109>

o Rotavirus: <https://doi.org/10.1073/pnas.1518977113>, <https://doi.org/10.1073/pnas.1719579115>

o Shigellosis: <https://doi.org/10.1371/journal.pone.0107223>

Responses: We express our gratitude to Reviewer #2 for their valuable comment and for sharing these references. We agree that different enteric pathogens may have different sensitivity to

climate-related factors and that a focus on diarrhea integrates over much underlying complexity. Although we were unable to study the dynamics of different pathogens without detailed pathogen-specific testing of stool specimens for every child in the study, we revised the Discussion section to acknowledge this nuance and motivate our focus on climate-sensitivity of the integrated clinical outcome, diarrhea, which still is of great public health and programmatic importance.

Lines 335-342: *“Furthermore, in this paper, we did not consider the various enteric pathogens and illnesses responsible for diarrhea, such as cholera, shigellosis, and rotavirus. These pathogens might exhibit distinct mechanisms and varying sensitivities to climate.³⁴⁻³⁸ The higher impact of WASH during the monsoon season on the risk of diarrhea might be attributed to the distinct impact of WASH on seasonally varying pathogens^{32,39}, whose transmission may not be solely driven by climatic conditions. Nevertheless, a study revealed that cholera and shigellosis, both identified as climate-sensitive, exhibited comparable relationships with climate variables despite differences in disease epidemiology.³⁷”*

Added the References:

32. Grembi, J. A. *et al.* Effect of Water, Sanitation, Handwashing, and Nutrition Interventions on Enteropathogens in Children 14 Months Old: A Cluster-Randomized Controlled Trial in Rural Bangladesh. *J. Infect. Dis.* **227**, 434–447 (2023).

38. Grembi, J. A. *et al.* Influence of climatic and environmental risk factors on child diarrhea and enteropathogen infection and predictions under climate change in rural Bangladesh. <http://medrxiv.org/lookup/doi/10.1101/2022.09.26.22280367> (2022)
doi:10.1101/2022.09.26.22280367.

39. Mertens, A. *et al.* Effects of water, sanitation, and hygiene interventions on detection of enteropathogens and host-specific faecal markers in the environment: a systematic review and individual participant data meta-analysis. *Lancet Planet. Health* **7**, e197–e208 (2023).

-On a similar note, different enteric diseases have different seasonal trends, and their association with temperature or flooding is not always monotonic. The result that the authors present here about a higher impact of WASH during the monsoon season on diarrheal risk might be due to a differential effect of WASH on different pathogens and not necessarily due to climatic conditions per se. Additionally, a previous work has already looked at the role of sanitation and socioeconomic variables on the spatial variation of rotavirus seasonality in Dhaka, finding that drinking water from tube wells, and not socioeconomic factors, has a protective effect against cases during the monsoon (<https://doi.org/10.1093/infdis/jiz436>).

Responses: We agree that such results might be partially influenced to the differential effect of WASH on different pathogens and not directly due to climate patterns. Indeed, seasonally-varying differences in pathogen transmission that align with changes in climate is a likely underlying mechanism to explain seasonal differences in WASH intervention effectiveness. We acknowledge this and have included this in the limitations.

Lines 338-340: *“The higher impact of WASH during the monsoon season on the risk of diarrhea might be attributed to the distinct impact of WASH on seasonally varying pathogens^{32,39}, whose transmission may not be solely driven by climatic conditions.”*

In the previous study cited by Reviewer #2, they used a different socioeconomic indicator – specifically father’s income and type of roofing – which is different from what we used in our present study. Additionally, the cited study was conducted in an urban context, contrasting with the

rural setting of our analysis. Nevertheless, our findings align reasonably well with theirs, as we observed a marginally statistically non-significant effect modification by wealth. It's noteworthy that across various wealth levels, seasons, and in combination with wealth and season, our effect modification analyses consistently demonstrated the protective nature or beneficial effects of WASH interventions compared to the control group.

- When the authors write, “which is consistent with a previous in-depth analysis of climate-related effect modifiers.”⁷ The reference is <https://doi.org/10.1101/2022.09.25.22280229>. “Influence of temperature and precipitation on the effectiveness of water, sanitation, and handwashing interventions against childhood diarrheal disease in rural Bangladesh: a re-analysis of a randomized control trial.” How is that previous preprint different from what is being presented here?

Responses: We thank Reviewer #2 for their comment. The focus of the preprint by Nguyen, A. et al. was on precipitation, temperature and seasonality acting as effect modifiers. In contrast, the present study centered on the modification of effects of WASH on diarrhea by wealth, as well as the combined effect by wealth and monsoon season. In this work, which we consider an extension of Nguyen et al., we focused on monsoon season as a composite measure of seasonal changes in climate indicators based on the findings of the climate indicator-specific analyses. Nguyen et al. also found that season appeared to be a strong effect modifier than temperature or precipitation alone. Consequently, evaluating the modification of effects by monsoon season alone was an essential preliminary step to comprehend how monsoon season might alter the impact of WASH on child diarrhea, before examining it in conjunction with wealth.

- “We estimated that a similar intervention at scale could prevent 734 cases per 1,000 children per month during the seasonal monsoon, with marked heterogeneity by region. The analysis demonstrates how to extend large-scale trials to inform WASH strategies among climate-sensitive and low-income populations.” I don't think two years of data is sufficient to make this conclusion. It oversimplifies the effect of climate covariates, their inter-annual variations, and the nonlinear nature of disease transmission.

Responses: We thank Reviewer #2 for this comment. We agree that relying solely on two years of data could potentially oversimplify the effects of climate covariates and their inter-annual variations. Nevertheless, it is extremely rare for RCTs to include more than 2 years of follow-up. Furthermore, this analysis included children from 360 communities throughout a large region of Bangladesh, thus capturing significant spatial heterogeneity within the time series. So, within the scope of extending inference from RCTs we believe the results represent a reasonable extension within the limits of the information available about treatment effects from an RCT. In principle, future studies could take a purely observational analysis approach to the question, but we are concerned about inference from observational analyses of WASH related exposures based on an internal analysis from three sister trials that included this trial from Bangladesh, which demonstrated unmeasured confounding that could only be controlled for through randomization, at least in the context of the growth endpoints ([https://www.thelancet.com/journals/langlo/article/PIIS2214-109X\(18\)30229-8/fulltext](https://www.thelancet.com/journals/langlo/article/PIIS2214-109X(18)30229-8/fulltext)). A more comprehensive and in-depth analysis over an extended period might provide a clearer understanding of the interplay between these variables and the efficacy of the interventions, especially within climate-sensitive and low-income populations, but whether that would be possible within a future RCT is unknown. One path forward might be the use of meta-analyses to pool information over multiple trials and thus capture more heterogeneity. We have put this in the limitations.

Lines 342-350: *“One last caveat to consider is that the data from two years may not have adequately encompassed the full scope of climate variability and inter-annual fluctuations, as well as the*

complex, non-linear dynamics of pathogen dynamics that cause diarrheal disease. Future studies over longer time periods could provide additional evidence for varying efficacy of WASH interventions by wealth and season, though randomized controlled trials of WASH interventions rarely extend for more than two years. Future meta-analyses that pool results over many different trials could help capture effect heterogeneity over a wider range of seasonally varying climate conditions.”

Reviewer #3 (Remarks to the Author):

What are the noteworthy results?

This interesting study uses data from a large-scale RCT in Bangladesh to do several things: 1) using pre-specified analysis to analyse the effect of combined WASH and combined WASH+nutrition interventions on reported diarrhoea in the diarrhoea-peak season (the monsoon); 2) using pre-specified analysis to estimate the effects of the intervention by wealth tertile; and 3) model the likely effects of the interventions across the country (in post-hoc analysis?). I would like to see this innovative paper published, provided the comments can be addressed, and I believe Nature Communications to be a relevant and appropriate outlet for it.

Responses: We are grateful to Reviewer 3 for their encouraging comment and support.

- Will the work be of significance to the field and related fields? How does it compare to the established literature? If the work is not original, please provide relevant references.

The work is potentially highly policy relevant, since it presents analysis of the time of year when WASH interventions can be most effective and also suggests that WASH is a highly pro-poor intervention, benefiting the poorest groups the most. Both are empirical questions that help resolve different competing theories about disease transmission, and the findings presented by the authors are also supported by a recent systematic review and meta-analysis of WASH interventions and child mortality cited by the authors. The paper also presents results of a method to estimate the likely effects of the intervention if it were possible to scale it up to the country level, by applying the estimated relationship between predictive factors and diarrhoea morbidity to geospatial data. The paper presents the implications of the findings in relevant formats, using estimated coefficients and figures to show the relationship between exposures and health inequality. . I wondered if it could be clearer what the WASH starting points were for the diarrhoea mapping exercise, and whether there was some way of making this clearer in the figure or associated text.

Responses: We express our gratitude to Reviewer #3 for their motivating comment. Concerning the mapping, we utilized a Generalized Additive Model (GAM) to estimate the impact of combined WASH interventions (a binary variable) on child diarrhea. The intervention arm served as the reference point for the WASH intervention variable. The assumption was that the control group reflects the baseline WASH conditions for the projection.

We clarified this in the Methods:

Lines 514-515: *“We assumed that the control group reflects the baseline WASH conditions for the projection.”*

Another implication of the analysis which I thought might be presented more clearly was the sensitivity analysis for the non-linear correlates of diarrhoea disease with respect to by mother's education; specifically, I wondered what the number of years of education was at each inflection point (the maximum and the minimum) and whether either of these were associated with completion of primary education, which itself is potentially policy-relevant since in Bangladesh great efforts have been made to get more girls into secondary education in recent decades which are

thought closely related to child health (e.g.

https://ieg.worldbankgroup.org/sites/default/files/Data/reports/impact_evaluation_bangladesh_child_health.pdf).

Responses: We provided this in the Supplementary Material (Supplementary Fig. 4) where we showed the non-linear relationship between diarrhea prevalence across the mother's educational rank based on the years of maternal education. Regarding the point about the education years completed, we added a figure to show diarrhea prevalence across different years of completed education (Supplementary Fig. 5).

Supplementary Fig. 5: Effect of WASH interventions on diarrhea by mother's educational years completed. **A:** Diarrhea prevalence across years of completed education in the control and intervention groups. **B:** Prevalence ratio of child diarrhea across years of completed education in the control and intervention groups. **C:** Prevalence difference of child diarrhea across years of completed education in the control and intervention groups. Shaded areas represent 95% confidence intervals.

- Does the work support the conclusions and claims, or is additional evidence needed?

I believe the work supports the findings and conclusions drawn, although I have a few queries about

the methods and results, as indicated below, which in my view would need to be resolved or addressed before publication.

Responses: We thank Reviewer #3 for their constructive and encouraging comments and suggestions.

- Are there any flaws in the data analysis, interpretation and conclusions? - Do these prohibit publication or require revision?

Can the authors respond to the following comments, which I believe should be addressed prior to publication:

- Throughout the paper, the authors refer to 'improved WASH' or the 'WASH intervention'. I think it should be clearer in the paper that the original trial on which this secondary analysis is based provided some aspects of 'improved WASH' but not all - notably I understand there was no intervention to provide improved water supply in quantity, which might itself also have an impact on diarrhoea disease (see Sharma-Waddington et al., 2023, PLOS Med).

Responses: We are grateful to Reviewer #3 for providing this valuable perspective. It is true that the water intervention did not involve any improvements to the water supply in terms of quantity. As per the primary outcome analysis performed by Luby, S. et al., the water intervention did not demonstrate a significant influence on child diarrhea.² To ensure clarity regarding the various intervention arms, we included the specifics of each intervention in the Methods section.

Added to the Methods section:

Lines 394-403: "The WASH Benefits Bangladesh trial comprised seven arms: water (W), sanitation (S), handwashing (H), nutrition (N), combined WSH, combined WSH + N, and double-sized control. The water intervention encompassed the use of an insulated storage container for drinking water, along with the utilization of Aquatabs. Sanitation efforts involved the implementation of a sani-scoop, potty, and an improved double-pit pour flush latrine. Handwashing interventions included the provision of a handwashing station, a storage bottle for soapy water, and laundry detergent sachets for the preparation of soapy water. Additionally, the nutrition arm entailed the use of lipid nutrient supplements (LNS) between 6-24 months, along with a storage container for LNS, exclusive breastfeeding until 180 days, and the introduction of complementary food at 6 months."

It is also not clear what the nutrition (N) component of the intervention was, which may be important since the authors argue later in the paper that they ensured consistency of the interventions analysed (combined WASH and WASH+N together), but it is not clear whether the nutrition intervention may have had theoretical or empirical effects on diarrhoea incidence (e.g. directly by providing clean food or indirectly by promoting greater resilience to infection through better nutritional status) over and above that from "improved WASH" alone.

Responses: As stated previously, we incorporated details about the nutrition interventions in the Methods section. The results of the primary outcome analysis by Luby, S. et al also showed that the nutrition intervention resulted in a noteworthy decrease in child diarrhea.² The authors speculated that this reduction might be attributed to improvements in breastfeeding practices or improvements in essential fatty acids and micronutrient status resulting from the intervention, potentially contributing to an improved gut epithelial immune response and subsequently reducing diarrhea.

² Luby, S. P. *et al.* Effects of water quality, sanitation, handwashing, and nutritional interventions on diarrhoea and child growth in rural Bangladesh: a cluster randomised controlled trial. *Lancet Glob. Health* 6, e302–e315 (2018).

Moreover, the research unveiled that the impacts of single sanitation, handwashing, combined WSH, and combined WSH + N were nearly equivalent in scale. Consequently, WSH+N did not provide substantial additional benefits in reducing diarrhea compared to WSH¹ which we mentioned in Lines 407-408:

"We found no evidence for any added benefit of combined WSH + N with respect to combined WSH in diarrhea.⁶"

- On the asset index, which is interpreted explicitly by the authors as representing wealth or socioeconomic position, the authors state that they excluded the components of the asset index often included in such indices relating to water and sanitation access, as argued by a UNICEF technical report. I do not disagree, but can it be clearer what the theoretical reason is for excluding these factors from the index. For example, if the reason is that, in a study examining correlates of diarrhoea outcomes, the wealth index should not include any factors which might have a direct effect on the outcome (since, in this analysis, they would not primarily capture variation in the wealth construct, but rather likely disease exposure), then it is not clear why the index also includes quality of flooring, or perhaps electricity or refrigerator, which also might be thought to affect infectious disease exposure in childhood directly.[There is also a policy angle to this, since analysis of the specific amenities available to households that affect their children's disease incidence directly may be policy actionable (e.g. flooring quality, rural electrification), whereas wealth usually isn't seen to be manipulable by policymakers, at least in the short term.]

Responses: To preserve the consistency assumption, we did not include WASH-related asset variables in the construction of the wealth index because of their correlation with the exposure (i.e., WASH interventions). We agree with Reviewer #3, and this finding suggests an alternative approach to enhancing population health in cases where intervening directly on wealth might be challenging. It signifies that we can assist vulnerable populations in reducing diarrhea and simultaneously enhance their resilience during the monsoon season by improving their WASH infrastructure and practices.

The authors also note that "we excluded asset based variables with near-to-zero variation and high levels of missingness" - can they clarify if this was why the radio and clock variables were excluded from the index? On the interpretation of the loading factors, only one of which is negative for the 'chouki'/stool, is this because households owning choukis were less likely to own chairs?

Responses: We are thankful to Reviewer #3 for this valuable feedback. We did, in fact, exclude the radio and clock variables due to their minimal variation. Regarding the negative factor loading for 'chouki'/stool, a potential explanation could be that more affluent households commonly possess chairs in the Western style, while the utilization of 'chouki' is associated with manual labor that requires squatting. We appreciate Reviewer #3's insightful input, and indeed, the negative value could also be attributed to the likelihood that wealthier households opt for chairs rather than choukis.

We added these points in the Methods section:

Lines 423-425: *"We excluded asset-based variables with near-to-zero variation (i.e., owning a radio and improved roof) and high levels of missingness (at least 10%, i.e., owning a clock).^{42,43}"*

Lines 435-442: *"Based on the factor loadings, some of the asset-based variables that contribute the most to the wealth variation are having one or more 'khat' or bed, electricity in the household, owning a TV and having one or more chair. In the meantime, the presence of a 'chouki,' a specific type of stool, demonstrates a negative factor loading, suggesting that wealthier households tend to have Western-style chairs, whereas the utilization of 'choukis' is linked to manual labor involving*

squatting. Another possible explanation could be the likelihood that wealthier households simply prefer chairs over 'choukis.'"

- When projecting the likely intervention effects country-wide, the authors state that they used WorldPop data based on the 2011 DHS survey wealth index "which is similar to the wealth index we used in the main analysis". If I understand what the authors have done correctly - they are using a method similar to small area estimation used in the poverty mapping literature - one of the assumptions of such analyses is that the variables used in both data sets should be the same or as similar as possible. With regards the wealth measure, it should be clearer how the two measures of wealth differed (presumably the WorldPop includes water and sanitation access, but anything else?), what the implications for (and limitations of) those differences might be. Depending on the answer to this question, the authors may also want to either present results of sensitivity analysis where they re-estimate their results using exactly the same wealth measure as the DHS 2011 index, so presumably including the WASH exposure measures, or re-estimate the WorldPop data using the wealth index used in the author's primary analysis, or at the very least present the loading factors for each index and the correlations across households of the two indices.

Responses: We express our gratitude to Reviewer #3 for this insightful feedback. In order to assess the similarity between the WorldPop-projected wealth index layer and the observed wealth index from the WASH Benefits Bangladesh trial, we computed the Pearson correlation and created a scatter plot (please refer to the figure below). Using the wealth rank, the observed correlation coefficient was approximately 0.39, indicating a considerable level of correlation between the completely independent wealth assessments. Meanwhile, when using the raw wealth scores, the correlation coefficient was about 0.43. We noted that this level of correlation was very similar as a recent study in *Science* that found correlations between 0.3-0.4 between a predicted wealth surface derived from satellite imagery and observed, survey-based measures of consumption.³ In our analysis based on the wealth rank, we adopted a more conservative approach. We agree with Reviewer #3 that this point should be mentioned in the manuscript.

We added in the Results section:

Lines 198-209: *"To evaluate the similarity between the WorldPop-projected wealth index layer and the observed wealth index from the WASH Benefits Bangladesh trial, we calculated the Pearson correlation and produced a scatter plot (Supplementary Fig. 3). The correlation between wealth measures was 0.39 (using wealth rank) and 0.43 (using wealth scores), suggesting a moderate degree of correlation, and of similar magnitude to other comparisons of national-level, projected wealth surfaces with directly measured wealth indices.²⁰ In our analysis, we adopted a more conservative approach by utilizing the wealth rank. However, it is important to highlight that variations exist in the construction of the wealth indices. The WorldPop wealth index, derived from the DHS, integrated WASH-related asset-based variables in its construction, whereas we opted to exclude them when constructing the wealth index for the WASH Benefits Bangladesh trial. Hence, it is essential to exercise caution when interpreting the projections in this context."*

³ Jean, N. *et al.* Combining satellite imagery and machine learning to predict poverty. *Science* **353**, 790–794 (2016).

Supplementary Fig. 3. Wealth rank and wealth scores based on the WorldPop predicted wealth index per grid-cell compared to block-level mean wealth rank based on the wealth index and block-level mean wealth scores measured from the WASH Benefits Bangladesh trial. The correlation was evaluated using the Spearman correlation coefficient. The fitted line was generated using a loess function.

We added in the References:

20. Jean, N. *et al.* Combining satellite imagery and machine learning to predict poverty. *Science* **353**, 790–794 (2016).

- Relatedly, also for the diarrhoea mapping exercise, can it be clearer what specific variables were used to predict diarrhoea by geographical area, and whether these included measures of existing water supplies, sanitation and hygiene exposures? Where the authors state that they only included geographies with "similar geographic patterns observed as the main analysis that focused on the monsoon period", what were these patterns?

Responses: We express our appreciation to Reviewer #3 for this valuable comment. Our intention behind mentioning similar geographic patterns was to specify that we restricted the projections to rural areas which is similar to the WASH Benefits Bangladesh trial's locations in four rural districts and included only grid cells with more than 2 children under 3 years indexed by the same wealth rank as the trial participants. We did not include measures of existing water supplies, sanitation and hygiene exposures.

The procedures we followed were as outlined below:

- 1) Used the WorldPop wealth index surface based on the 2011 DHS survey's wealth index.
- 2) Utilized WorldPop to estimate the number of children under 3 years in each grid cell at a 1-kilometer resolution.
- 3) Restricted projections to rural areas of Bangladesh by masking urban areas, as determined by the Global Human Settlement.
- 4) Used GAM model prevalence difference estimates between the control and WASH based on wealth index (specifically wealth rank) to project diarrhea cases by integrating steps 1 to 4.
- 5) Applied the GAM model for WASH diarrhea prevalence difference between control and WASH arms by wealth rank to estimate the number of prevented cases by multiplying it by the proportion of children under 3 years indexed by wealth rank during the rainy season.

$$diarrheapreventedbyWASH = childrenunder3y \times PD(control - WASH)_{i.monsoon.rural} \times 4 weeks$$

To clarify, we added the following to the main text in the Methods section:

Lines 508-518: *“First, we used the WorldPop projected wealth index layer based on the 2011 DHS wealth index. Second, we used WorldPop population layer to estimate the number of children under 3 years in each grid-cell at a 1-kilometer resolution. Third, we restricted the projections to rural areas of Bangladesh by masking urban areas defined using the Global Human Settlement. Fourth, we used GAM model prevalence difference estimates between the control and WASH based on wealth index (specifically wealth rank) to project diarrhea cases by integrating steps 1 to 4. We assumed that the control group reflects the baseline WASH conditions for the projection. Fifth, we applied the GAM model for diarrhea prevalence difference between control and WASH arms by wealth rank to estimate the number of prevented cases by multiplying it by the proportion of children under 3 years indexed by wealth rank during the monsoon season.”*

Regarding the second query, we understand that Reviewer #3 was alluding to lines 222-227, where we performed a sensitivity analysis projecting the wealth-based reduction in diarrhea cases averaged throughout the year, not exclusively during the monsoon season. The geographical patterns we discussed pertained to the regions that exhibited the most significant benefits, specifically the areas in north-central and north-west Bangladesh. These were the same geographic areas that demonstrated the greatest benefits in the primary analysis when considering only the monsoon season.

We clarified this in the Results section:

Lines 237-243: *“Furthermore, we projected the diarrhea cases prevented by wealth across the entire year due to an efficacious WASH was 339 cases (95% CI 105, 573) per 1,000 children under 3 years per month. This analysis yielded results consistent with the primary findings, revealing similar geographical patterns to those observed in the main analysis concentrated on the monsoon period. Notably, the regions in north-central and north-west Bangladesh emerged as the regions that would benefit the most, although the overall effect size was slightly reduced when averaged over the whole year (Supplementary Fig. 7).”*

- A minor point, where the authors calculated the total number of children under 3 years by multiplying the total number of under-14s by 3/14, is this an accurate calculation based on the expected age distributions of children in Bangladesh?

Responses: We assumed a uniform age distribution within the 0-14y range based on the WorldPop population layer, thus employing the proportion of 0.21 or 3/14. To confirm this, we utilized data from the Demographic and Health Survey for Bangladesh. We demonstrate that the age distribution within this age range is fairly uniform, with children under 3 years old making up about 19% (please refer to the figure below), an estimate that is quite close to 21%.

Prior to this, we estimated the proportion of children between 0-14y by multiplying the total population count per grid-cell by 0.31, an approximation of the proportion of children between 0-14 years provided by the World Bank for Bangladesh.

We added this point in the Methods section:

Lines 457-458: *“We assumed a uniform age distribution within the 0-14 range, thus employing the proportion of 0.21.”*

-Is the methodology sound? Does the work meet the expected standards in your field?

I believe the study uses high quality methods to generate innovative policy-relevant findings, although what was done could be reported more clearly, and more clarity on the differences between the two measures of wealth used in the paper should be given, as indicated above.

Responses: We thank Reviewer #3 for their encouraging comment. We hope that the changes we made adequately addressed their concerns.

- Is there enough detail provided in the methods for the work to be reproduced?

It is generally clear, although what was done for the diarrhoea mapping exercise could be reported more clearly, as indicated above.

Responses: We appreciate Reviewer #3's positive feedback. As indicated in our previous responses, we have made the suggested changes to enhance the clarity of the mapping analysis.

Thank you for considering our manuscript.

Yours sincerely,

Pearl Anne Ante-Testard, PhD, on behalf of all authors
Postdoctoral Scholar
F.I. Proctor Foundation,
University of California, San Francisco

REVIEWERS' COMMENTS

Reviewer #1 (Remarks to the Author):

Thank you for the thorough handling of my comments, which have about a 10% "hallucination" rate, about the same I find with ChatGPT-4.

In your future endeavors, you probably should avoid the term "overlapping CIs" in most situations, as two 95% CIs can overlap by over 25% of their length, and still be statistically significantly different at the 0.05 level (triangle inequality). I am not a stickler regarding the 0.05 level, but I really need more evidence before causality can be decided, as is in the phrase "the WASH intervention marginally decreased". I'm sure SLuby has a rule against using such phraseology. You can of course say something (a point estimate) was lower, because it was, but assigning causality is to be avoided.

Great to see the regression methods unified and better explained. Thanks for the Yelland ref, which on its own was really not a very useful article, although fine to reference here. The very nature of the sandwich estimator is that it will track the true variance at the level it is applied and all lower levels.

I took another look at the prevalence=incidence assumption, and it probably does not alter the basic findings of the manuscript. The calculation of x4 instead of 4.3 underestimates incidence by about 8%, and ignoring factoring in longer diarrhea episodes overestimates incidence by about the same magnitude—so, on average, you're probably not far off. And since averages (totals) are the main thing here, you shouldn't suffer the fate of the statistician who drowned wading across the river that was an average 2 feet deep.

That said, shame on you for not doing your own calculation: "cohort in Bangladesh that found 1350 out of 1566 (88.5%) of diarrhea episodes" I think you'll find 1350/1526 is closer to 88.5%-- and for good reason.

Reviewer #2 (Remarks to the Author):

While I acknowledge that the findings of WASH intervention have a more substantial impact on lower socioeconomic groups, I still have some concerns about this revised version.

Authors from this group have already published an article in 2022 (<https://doi.org/10.1371/journal.pmed.1004041>), where they analyzed the data from this study, finding that the effect of the intervention was only seen during monsoon, which is redundant with findings from figure 4 and 5. The authors also have another preprint (<https://doi.org/10.1101/2022.09.25.22280229>) where they explicitly look at the effect of temperature and precipitation, finding that the WASH interventions are more effective in preventing diarrhea during monsoon season, especially when this is followed heavy rain and high temperatures. The overlap is substantial among these three studies and is not fully acknowledged in the revised text.

In addition, the authors did a linear extrapolation of the cases prevented by WASH, ignoring the complexity of the interaction between diarrheal diseases, the monsoon seasons and their inter annual variations. With only two years of data from the WASH and in the light of extensive literature showing the association between climate (and monsoon in particular) and transmission of infectious diseases, the results presented in Figure 6 are very difficult to validate.

Reviewer #3 (Remarks to the Author):

Thanks for addressing my comments. I have no further comments. I do feel that a more rigorous analysis for research question 3 would have used the same index for both surveys, but the authors now transparently report the correlation between indices and a supportive reference.